# Secure Network Release with Link Privacy

## Abstract

Many data mining and analytical tasks rely on the abstraction of networks (graphs) to summarize relational structures among individuals (nodes). Since relational data are often sensitive, we aim to seek effective approaches to release utility-preserved yet privacy-protected structured data. In this paper, we leverage the differential privacy (DP) framework to formulate and enforce rigorous privacy constraints on deep graph generation models, with a focus on edge-DP to guarantee individual link privacy. In particular, we enforce edge-DP by injecting Gaussian noise to the gradients of a link reconstruction based graph generation model, while ensuring data utility by improving structure learning with structure-oriented graph comparison. Extensive experiments on two real-world network datasets show that our proposed DPGGAN model is able to generate networks with effectively preserved global structure and rigorously protected individual link privacy.

## 1 Introduction

Nowadays, open data of networks play a pivotal role in data mining and data analytics (Tang et al., 2008; Sen et al., 2008; Blum et al., 2013; Leskovec & Krevl, 2014). By releasing and sharing structured relational data with research facilities and enterprise partners, data companies harvest the enormous potential value from their data, which benefits decision-making on various aspects, including social, financial, environmental, through collectively improved ads, recommendation, retention, and so on (Yang et al., 2017; 2018; Sigurbjörnsson & Van Zwol, 2008; Kuhn, 2009). However, network data usually encode sensitive information not only about individuals but also their interactions, which makes direct release and exploitation rather unsafe. More importantly, even with careful anonymization, individual privacy is still at stake under collective attack models facilitated by the underlying network structure (Zhang et al., 2019; Cai et al., 2018). Can we find a way to securely release network data without drastic sanitization that essentially renders the released data useless?

In dealing with such tension between the need to release utilizable data and the concern of data owners' privacy, quite a few models have been proposed recently, focusing on grid-based data like images, texts and gene sequences (Frigerio et al., 2019; Papernot et al., 2018; Triastcyn & Faltings, 2018; Narayanan & Shmatikov, 2008; Xie et al., 2018; Chen et al., 2018; Boob et al., 2018; Dy & Krause, 2018; Lecuyer et al., 2018; Zhang et al., 2018). However, none of the existing models can be directly applied to the network (graph) setting. While a secure generative model on grid-based data apparently aims to preserve high-level semantics (*e.g.*, class distributions) and protect detailed training data (*e.g.*, exact images or sentences), it remains obtuse what to be preserved and what to be protected for network data, due to its modeling of complex interactive objects.

**Motivating scenario.** In Figure 1, a bank aims to encourage public studies on its customers' community structures. It does so by firstly anonymizing all customers and then sharing the network (*i.e.*, (a) in Figure 1) to the public. However, an attacker interested in knowing the financial interactions (*e.g.*, money transfer) between particular customers in the bank may happen to have access to another network of a similar set of customers (*e.g.*, a malicious employee of another financial company). The similarity of simple graph properties like node degree distribution and triangle count between the two networks can then be used to identify specific customers with high accuracy in the released network (*e.g.*, customer $A$ as the only node with degree 5 and within 1 triangle, and customer $B$ as the only node with degree 2 and within 1 triangle). Thus, the attacker confidently knows the A and B's identities and the fact that they have financial interactions in the bank, which seriously harms customers' privacy and poses potential crises.

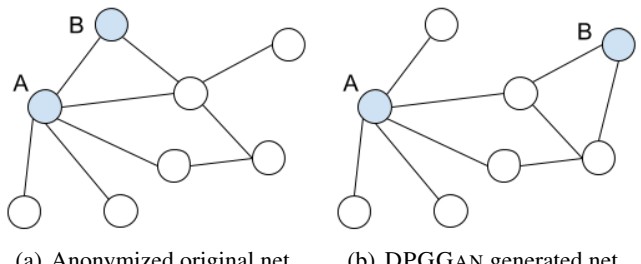

(a) Anonymized original net.    (b) DPGGAN generated net.

Figure 1: **A toy pair of anonymized and generated networks.**

As the first contribution in this work, we define and formulate secure network release goals as *preserving global network structure* while *protecting individual link privacy*. Continue with the toy example, the solution we propose is to train a graph neural network model on the original network and release the generated networks (*e.g.*, (b) in Figure 1). Towards the utility of generated networks, we require them to be similar to the original networks from a global perspective, which can be measured by various graph global properties (*e.g.*, network (b) has very similar degree distribution and the same triangle count as (a)). In this way, we expect many downstream data-mining and analytical tasks on them to produce similar results as on the original networks. As for privacy protection, we require that the information in the generated networks cannot confidently reveal the existence or absence of any individual links in the original networks (*e.g.*, the attacker may still identify customers A and B in network (b), but their link structure has changed).

Subsequently, there are two unique challenges in learning such structure-preserved and privacy-protected graph generation models, which have not been explored by existing literature so far.

**Challenge 1: Rigorous protection of individual link privacy.** The rich relational structures in graph data often allow attackers to recover private information through various ways of collective inference (Zhang et al., 2014; Narayanan & Shmatikov, 2009; Backstrom et al., 2007). Moreover, graph structure can always be converted to numerical features such as spectral embedding, after which most attacks on grid-based data like model inversion (Fredrikson et al., 2015) and membership inference (Shokri et al., 2017) can be directly applied for link identification. How can we design an effective mechanism with rigorous privacy protection on links in networks against various attacks?

**Challenge 2: Effective preservation of global network structure.** To capture the global network structure, the model has to constantly compare the structures of the input graphs and currently generated graphs during training. However, unlike images and other grid-based data, graphs have flexible structures, and thus they lack efficient universal representations (Dong et al., 2019). How can we allow a network generation model to effectively learn from the structural difference between two graphs, without conducting very time-costly operations like isomorphism tests all the time?

**Present work.** In this work, for the first time, we draw attention to the secure release of network data with deep generative models. Technically, towards the aforementioned two challenges, we develop Differentially Private Graph Generative Nets (DPGGAN), which imposes DP training over a link reconstruction based network generation model for rigorous individual link privacy protection, and further ensures structure-oriented graph comparison for effective global network structure preservation. In particular, we first formulate and enforce edge-DP via Gaussian gradient distortion by injecting designed noise into the sensitive modules during model training. Then we leverage graph convolutional networks (Kipf & Welling, 2017) through a variational generative adversarial network architecture (Gu et al., 2019; Larsen et al., 2016) to enable structure-oriented network comparison.

To evaluate the effectiveness of DPGGAN, we conduct extensive experiments on two real-world network datasets. On one hand, we evaluate the utility of generated networks by computing a suite of commonly concerned graph properties to compare the global structure of generated networks with the original ones. On the other hand, we validate the privacy of individual links by evaluating links predicted from the generated networks on the original networks. Consistent experimental results show that DPGGAN is able to effectively generate networks that are similar to the original ones regarding global network structure, while at the same time useless towards individual link prediction.

## 2 RELATED WORK

**Differential Privacy (DP).** Differential privacy is a statistical approach in addressing the paradox of learning nothing about an individual while learning useful information about a population (Dwork et al., 2006). Recent advances in deep learning have led to the rapid development of DP-oriented learning schemes. Among them, the Gaussian Mechanism (Dwork et al., 2014), defined as follows, provides a neat and compatible framework for DP analysis over machine learning models.

**Definition 1** (Gaussian Mechanism (Dwork et al., 2014)). *For a deterministic function $f$ with its $\ell_2$-norm sensitivity as $\Delta_2 f = \max_{\|\mathbf{G}-\mathbf{G}'\|_1=1} \|f(\mathbf{G}) - f(\mathbf{G}')\|_2$, we have:*

$$\mathcal{M}_f(\mathbf{G}) \triangleq f(\mathbf{G}) + \mathcal{N}(0, \Delta_2 f^2 \sigma^2), \tag{1}$$

*where $\mathcal{N}(0, \Delta_2 f^2 \sigma^2)$ is a random variable obeying the Gaussian distribution with mean 0 and standard deviation $\Delta_2 f \sigma$. The randomized mechanism $\mathcal{M}_f(\mathbf{G})$ is $(\varepsilon, \delta)$-DP if $\sigma \geq \Delta_2 f \sqrt{2\ln(1.25/\delta)}/\varepsilon$ and $\varepsilon < 1$.*

Following this framework, (Abadi et al., 2016) proposes a general training strategy called DPSGD, which looses the condition on the overall privacy loss than that in Definition 1 by tracking detailed information of the SGD process to achieve an adaptive Gaussian Mechanism.

DP learning has also been widely adapted to generative models (Frigerio et al., 2019; Papernot et al., 2018; Triastcyn & Faltings, 2018; Narayanan & Shmatikov, 2008; Mohammed et al., 2011; Xie et al., 2018; Chen et al., 2018; Boob et al., 2018; Dy & Krause, 2018; Lecuyer et al., 2018; Zhang et al., 2018). For example, (Frigerio et al., 2019; Chen et al., 2018; Boob et al., 2018; Zhang et al., 2018) share the same spirit by enforcing DP on the discriminators, and thus inductively on the generators, in a generative adversarial network (GAN) scheme. However, none of them can be directly applied to graph data due to the lack of consideration of structure generation.

For graphs' structural data, two types of privacy constraints can be applied, *i.e.*, node-DP (Kasiviswanathan et al., 2013) and edge-DP (Blocki et al., 2012), which define two neighboring graphs to differ by at most one node or edge. In this work, we aim at the secure release of network data, and particularly, we focus on edge privacy because it is essential for the protection of object interactions unique for network data compared with other types of data. Several existing works have studied the protection of edge-DP. For example, (Sala et al., 2011) generates graphs based on the statistical representations extracted from the original graphs blurred by designed noise, whereas (Wang & Wu, 2013) enforces the parameters of dK-graph models to be private. However, based on shallow graph generation models, they do not flexibly capture global network structure that can support various unknown downstream analytical tasks (Zhang et al., 2019; Wasserman & Zhou, 2010).

**Graph Generation (GGen).** GGen has been studied for decades and is widely used to synthesize network data used to develop various collective analysis and mining models (Evans & Lambiotte, 2009; Hallac et al., 2017). Earlier works mainly use probabilistic models to generate graphs with certain properties (Erdős & Rényi, 1960; Watts & Strogatz, 1998; Barabási & Albert, 1999; Newman, 2001), which are manually designed based on sheer observations and prior assumptions.

Thanks to the surge of deep learning, many advanced GGen models have been developed recently, which leverage different powerful neural networks in a learn-to-generate manner (Kipf & Welling, 2016; Bojchevski et al., 2018; You et al., 2018b; Simonovsky & Komodakis, 2018; Li et al., 2018; You et al., 2018a; Jin et al., 2018; Grover et al., 2017; De Cao & Kipf, 2018; Zou & Lerman, 2018; Ma et al., 2018). For example, NetGAN (Bojchevski et al., 2018) converts graphs into biased random walks, learns the generation of walks with GAN, and assembles the generated walks into graphs; GraphRNN (You et al., 2018b) regards the generation of graphs as node-and-edge addition sequences, and models it with a heuristic breadth-first-search scheme and hierarchical RNN. These neural network based models can often generate graphs with much richer properties, and flexible structures learned from real-world graphs.

To the best of our knowledge, no existing work on deep GGen has looked into the potential privacy threats laid during the learning and releasing of the powerful models. Such concerns are rather urgent in the network setting, where sensitive information can often be more easily compromised in a collective manner (Dai et al., 2018; Backstrom et al., 2007; Zhang et al., 2014) and privacy leakage can easily further propagate (Narayanan & Shmatikov, 2009; Zügner et al., 2018).

## 3    DPGGAN

In this work, we propose DPGGAN for the secure release of generated networks, whose global graph structures are similar to the original sensitive networks, but the individual links (edges) between objects (nodes) are safely protected. To provide robust privacy guarantees towards various graph attacks, we propose to leverage the well-studied technique of differential privacy (DP) (Dwork et al., 2014) by enforcing the edge-DP defined as follows.

**Definition 2** (Edge Differential Privacy (Blocki et al., 2012)). *A randomized mechanism $\mathcal{M}$ satisfies $(\varepsilon, \delta)$-edge-DP if for any two neighboring graphs $\mathbf{G}_1, \mathbf{G}_2 \in \mathcal{G}$, which differ by at most one edge, $\Pr[\mathcal{M}(\mathbf{G}_1) \in S] \leq exp(\varepsilon) \times \Pr[\mathcal{M}(\mathbf{G}_2) \in S] + \delta$, where $S \subset range(\mathcal{M})$.*

Our key insight is, *a graph generation model $\mathcal{M}$ satisfying the above edge-DP should learn to generate similar graphs given the input of two neighboring graphs that differ by at most one edge; as a consequence, the information in the generated graph does not confidently reveal the existence or absence of any one particular edge in the original graph, thus protecting individual link privacy.*

To ensure DP on individual links, we exploit the existing link reconstruction based graph generation model GVAE (Kipf & Welling, 2016), and design a training algorithm to dynamically distort the gradients of its sensitive model parameters by injecting proper amounts of Gaussian noise based on the framework of DPSGD (Abadi et al., 2016). We provide theoretical analysis on applying DPSGD to achieve edge-DP with GVAE based on the nature of graph data and the link reconstruction loss. Moreover, to improve the capturing of global graph structures, we replace the direct binary cross-entropy (BCE) loss on graph adjacency matrices in GVAE with a structure-oriented graph discriminator based on GCN (Kipf & Welling, 2017) and the framework of VAEGAN (Gu et al., 2019; Larsen et al., 2016). We further prove the improved model to maintain the same edge-DP.

**Backbone GVAE.** Recent research on graph models has been primarily focused around GCN (Kipf & Welling, 2017), which is shown to be promising in calculating universal graph representations (Maron et al., 2019; Xu et al., 2019; Chen et al., 2019; Keriven & Peyré, 2019). In this work, we harness the power of GCN under the consideration of edge-DP by adapting the link reconstruction based graph variational autoencoder (GVAE) (Kipf & Welling, 2016) as our backbone graph generation model.

Notably, we are given a graph $\mathbf{G} = \{\mathbf{V}, \mathbf{E}\}$, where $\mathbf{V}$ is the set of $N$ nodes (vertices), and $\mathbf{E}$ is the set of $M$ links (edges), which can be further modeled by a binary adjacency matrix $\mathbf{A}$. As a common practice (Hamilton et al., 2017), we set the node features $\mathbf{X}$ simply as the one-hot node identity matrix. The autoencoder architecture of GVAE consists of a GCN-based graph encoder to guide the learning of a feedforward neural network (FNN) based adjacency matrix decoder, which can be trained to directly reconstruct graphs with similar links as in the input graphs. A stochastic latent variable $\mathbf{Z}$ is further introduced as the latent representation of $\mathbf{A}$ as

$$q(\mathbf{Z}|\mathbf{X}, \mathbf{A}) = \prod_{i=1}^{N} q(\mathbf{Z}_i|\mathbf{X}, \mathbf{A}) = \prod_{i}^{N} \mathcal{N}(\mathbf{z}_i|\mu_i, \mathrm{diag}(\sigma_i^2)), \qquad (2)$$

where $\mu = \mathbf{g}_\mu(\mathbf{X}, \mathbf{A})$ is the matrix of mean vectors $\mu_i$, and $\sigma = \mathbf{g}_\sigma(\mathbf{X}, \mathbf{A})$ is the matrix of standard deviation vectors $\sigma_i$. $\mathbf{g}_\bullet(\mathbf{X}, \mathbf{A}) = \tilde{\mathbf{A}}\mathrm{ReLU}(\tilde{\mathbf{A}}\mathbf{X}\mathbf{W}_0)\mathbf{W}_1$ is a two-layer GCN model. $\mathbf{g}_\mu$ and $\mathbf{g}_\sigma$ share the first-layer parameters $\mathbf{W}_0$. $\tilde{\mathbf{A}} = \mathbf{D}^{-\frac{1}{2}}\mathbf{A}\mathbf{D}^{-\frac{1}{2}}$ is the symmetrically normalized adjacency matrix of $\mathbf{G}$, with degree matrix $\mathbf{D}_{ii} = \sum_{j=1}^{N} \mathbf{A}_{ij}$. $\mathbf{g}_\mu$ and $\mathbf{g}_\sigma$ form the encoder network.

To generate a graph $\mathbf{G}'$, a reconstructed adjacency matrix $\mathbf{A}'$ is computed from $\mathbf{Z}$ by an FNN decoder

$$p(\mathbf{A}|\mathbf{Z}) = \prod_{i=1}^{N} \prod_{j=1}^{N} p(\mathbf{A}_{ij}|\mathbf{z}_i, \mathbf{z}_j) = \prod_{i=1}^{N} \prod_{j=1}^{N} \sigma(\mathbf{f}(\mathbf{z}_i)^T \mathbf{f}(\mathbf{z}_j)), \qquad (3)$$

where $\sigma(z) = 1/(1 + e^{-z})$, $\mathbf{f}$ is a two-layer FNN appended to $\mathbf{Z}$ before the logistic sigmoid function. The whole model is trained through optimizing the following variational lower bound

$$\mathcal{L}_{vae} = \mathcal{L}_{\mathrm{rec}} + \mathcal{L}_{\mathrm{prior}} \qquad (4)$$
$$= \mathbb{E}_{q(\mathbf{Z}|\mathbf{X}, \mathbf{A})}[\log p(\mathbf{A}|\mathbf{Z})] - D_{\mathrm{KL}}(q(\mathbf{Z}|\mathbf{X}, \mathbf{A})\|p(\mathbf{Z})),$$

where $\mathcal{L}_{\mathrm{rec}}$ is implemented as the sum of an element-wise binary cross entropy (BCE) loss between the adjacency matrices of the input and generated graphs, and $\mathcal{L}_{\mathrm{prior}}$ is a prior loss based on the Kullback-Leibler divergence towards the Gaussian prior $p(\mathbf{Z}) = \prod_{i=1}^{N} p(\mathbf{z}_i) = \prod_{i}^{N} \mathcal{N}(\mathbf{z}_i|\mathbf{0}, \mathbf{I})$.

**Enforcing DP.** The probabilistic nature of $\mathbf{Z}$ allows the model to be generative, meaning that after training the model with an input graph $\mathbf{G}$, we can detach and disregard the encoder, and then freely generate an unlimited amount of graphs $\mathbf{G}'$ with similar links to $\mathbf{G}$, by solely drawing random samples of $\mathbf{Z}$ from the prior distribution $\mathcal{N}(\mathbf{0}, \mathbf{I})$ and computing $\mathbf{A}'$ with the learned decoder network *w.r.t.* Eq. (3). However, as shown in (Kurakin et al., 2017; Gondim-Ribeiro et al., 2018), powerful neural network models like VAE can easily overfit training data, so directly releasing a trained GVAE model poses potential privacy threats, as links in its generated graphs may be highly indicative towards links in the training graphs.

In this work, we care about the generation model's rigorously protecting the privacy of individual links in the training data, *i.e.*, ensuring edge-DP. Particularly, in Definition 2, the inequality guarantees that the distinguishability of any one edge in the graph will be restricted to the privacy leak level proportional to $\varepsilon$, quantifying the absolute value of privacy information possibly to be leaked by a graph generation model.

According to Eq. (3), GVAE essentially takes a graph $\mathbf{G}$, in particular, the links $\mathbf{E}$ among the nodes $\mathbf{V}$ in $\mathbf{G}$, as input and generates a new graph $\mathbf{G}'$ by reconstructing the links $\mathbf{E}'$ among the same set of nodes $\mathbf{V}$. Therefore, if we regard GVAE as the mechanism $\mathcal{M}$, as long as its model parameters are properly randomized, the framework satisfies edge-DP. To be specific, any two input graphs $\mathbf{G}_1$ and $\mathbf{G}_2$ differing by at most one link in principle lead to similar generated graphs $\mathbf{G}'$, so information in $\mathbf{G}'$ does not confidently reveal the existence or absence of any particular link in $\mathbf{G}_1$ or $\mathbf{G}_2$.

To exploit the well-structured graph generation framework of GVAE, we leverage the Gaussian mechanism (Definition 1) (Dwork et al., 2014) and DPSGD (Abadi et al., 2016) to enforce edge-DP on it. In our setting, $\mathbf{G}$ is the original training graph. Then Eq. (1) tells us that a link reconstruction based graph generation model $\mathcal{M}$ can be randomized to ensure $(\varepsilon, \delta)$-edge-DP with properly parameterized Gaussian noise. Prominently, we follow DPSGD (Abadi et al., 2016) to inject a designed Gaussian noise to the gradients of our decoder network clipped by a hyper-parameter $C$ as follows.

$$\tilde{g}_{\theta, \mathcal{L}} = \frac{1}{N} \left( \sum_{i=1}^{N} \left( \nabla_{v_i, \theta} \mathcal{L} / \max(1, \frac{\|\nabla_{v_i, \theta} \mathcal{L}\|_2}{C}) \right) + \mathcal{N}(0, \sigma^2 C^2 \mathbf{I}) \right), \tag{5}$$

where $\mathcal{L}$ is the loss function of a link reconstruction based graph generation model, $C$ is the clipping hyper-parameter for the model's original gradient to bound the influence of each link, and $\sigma$ is the noise scale hyper-parameter. In the following theorem, we analyze and prove that the noised clipped gradient $\tilde{g}_{\theta, \mathcal{L}}$ applied as above guarantees the learned graph generation model to be edge-DP, with a different condition from that in Definition 1 due to the learning process of link based graph generation model.

**Theorem 1.** *In training a link reconstruction based graph generation model on a graph with $N$ nodes with batch size $B$, given the sampling probability $q = B/N$, and the number of steps $T$, there exist explicit constants $c_1$ and $c_2$ that for any $\varepsilon < c_1 q^2 T$, iteratively updating the model $T$ times with $\tilde{g}_{\theta, \mathcal{L}}$ attains it with $(\varepsilon, \delta)$-edge-DP for any $\delta > 0$ if we choose*

$$\sigma \geq c_2 \frac{q\sqrt{T \log(1/\delta)}}{\varepsilon},$$

*where $c_1 \geq \frac{1}{c_0} \log \frac{1}{q\sigma}$, $c_2 \leq 1/\sqrt{c_0(1 - c_0)}$ for any $c_0 \in (0, 1)$.*

The proofs of Theorem 1 are detailed in Appendix A.

For the training of the DPGVAE decoder, $\mathcal{L}$ in Eq. (5) is specified as $\mathcal{L}_{rec}$ in Eq. (4). Due to the link reconstruction nature of DPGVAE, we derive Corollary 1.1 from Theorem 1 as follows.

**Corollary 1.1** (DPGVAE edge-DP). *Under the same conditions in Theorem 1, iteratively updating the decoder in DPGVAE for $T$ times with $\tilde{g}_{\theta, \mathcal{L}_{rec}}$ attains it with $(\varepsilon, \delta)$-edge-DP.*

In the generation stage, we can disregard the encoder and only use the decoder to generate an unlimited amount of graphs from randomly sampled vectors from the prior distribution $\mathcal{N}(\mathbf{0}, \mathbf{I})$. Due to the randomness of the normal Gaussian distribution, the sampling process can be regarded as $(0, 0)$-DP. By the composability property of DP (Dwork et al., 2014), generating graphs from random noises with the DPGVAE decoder satisfies $(\varepsilon, \delta)$-edge-DP, whose release in principle does not disclose sensitive information regarding individual links in the original sensitive networks. Since we do not release the encoder network, we do not need to clip and perturb its gradients during training to induce minimum interruptions.

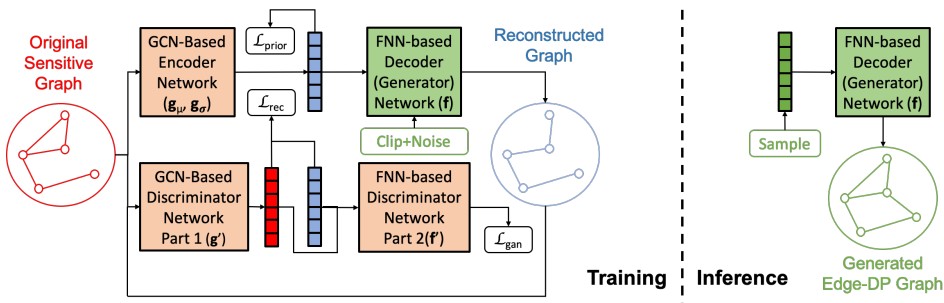

Figure 2: **Neural architecture of DPGGAN (best viewed in color)**: Our novel graph generation model consists of a GCN-based encoder, an FNN-based decoder (generator), and a GCN+FNN-based discriminator. Sensitive data and modules are marked as red, while safe operations (*i.e.*, gradient clipping, noise injection and sampling) are marked as green, leading to DP modules and data.

**Improving structure learning.** Besides individual link privacy, we also aim to preserve the global network structure to ensure the utility of released data. As we discuss before, original GVAE computes the reconstruction loss between input and generated graphs based on the element-wise BCE between their adjacency matrices. Such a computation is specified on each link, rather than the graph structure as a whole. To improve the global graph structure learning, we leverage GCN again, which has been shown universally powerful in capturing graph-level structures (Maron et al., 2019; Xu et al., 2019; Chen et al., 2019; Keriven & Peyré, 2019). In particular, we borrow the framework of VAEGAN from recent research (Gu et al., 2019; Larsen et al., 2016; Yang et al., 2019), and compute a structure-oriented generative adversarial network (GAN) loss as

$$\mathcal{L}_{gan} = \log(\mathcal{D}(\mathbf{A})) + \log(1 - \mathcal{D}(\mathbf{A}'))$$
$$\text{with } \mathcal{D}(\mathbf{A}) = \mathbf{f}'(\mathbf{g}'(\mathbf{X}, \mathbf{A})), \tag{6}$$

where $\mathbf{g}'$ and $\mathbf{f}'$ are GCN and FNN networks similar as defined before, besides at the end of $\mathbf{g}'$ the node-level representations are summed up as the graph-level representation, which resembles the recently proposed GIN model for graph-level representation learning (Xu et al., 2019). In this DPGGAN framework, the decoder also serves as the generator, while $\mathcal{D} = \mathbf{f}' \cdot \mathbf{g}'$ is the discriminator.

Following (Gu et al., 2019), the encoder is trained *w.r.t.* $\mathcal{L}_{rec} + \lambda_1 \mathcal{L}_{prior}$, the generator *w.r.t.* $\mathcal{L}_{rec} - \lambda_2 \mathcal{L}_{gan}$, and the discriminator *w.r.t.* $\lambda_2 \mathcal{L}_{gan}$, where $\lambda_1$ and $\lambda_2$ are hyper-parameters. To enforce DP constraints and complete our proposed DPGGAN framework, Eq. (5) with $\mathcal{L}_{rec}$ substituted by $\mathcal{L}_{rec} - \lambda_2 \mathcal{L}_{gan}$ is applied to distort the gradients of the generator and guarantee edge-DP, which can be used to securely generate networks with the other parts disregarded after training. The overall framework of DPGGAN is shown in Figure 2, and the training process is detailed in Appendix B.

The intuition behind the novel design of DPGGAN is, the GCN encodings $\mathbf{g}'(\mathbf{A})$ and $\mathbf{g}'(\mathbf{A}')$ capture the graph structures of $\mathbf{G}$ and $\mathbf{G}'$, so a reconstruction loss $\mathcal{L}_{rec} = \|\mathbf{g}'(\mathbf{A}) - \mathbf{g}'(\mathbf{A}')\|_2^2$ captures the intrinsic structural difference between $\mathbf{G}$ and $\mathbf{G}'$ instead of the simple sum of the differences over their individual links. Note that the effectiveness of our structure-oriented discriminator is critical not only because it can directly enforce effective training of the graph generator through the minimax game in Eq. (6), but also because it can learn to relax the penalty on certain individual links through flexible and diverse configurations of the whole graph as long as the global structures remain similar, which exactly fulfills our goals of secure network release. The benefits of such diversity enabled by the VAEGAN have also been discussed in image generation (Gu et al., 2019; Larsen et al., 2016).

Compared with DPGVAE, DPGGAN does not directly compute the link reconstruction loss based on BCE in Eq. (4), but rather computes it based on the graph discriminator $\mathcal{D}$. However, the link reconstruction based graph generator of DPGGAN is exactly the same as DPGVAE. Since we also do not release $\mathcal{D}$ after training, we can simply retrieve Corollary 1.2 from Theorem 1 as follows.

**Corollary 1.2** (DPGGAN edge-DP). *Under the same conditions in Theorem 1, iteratively updating the generator in DPGGAN for $T$ times with $\tilde{g}_{\theta, (\mathcal{L}_{rec} - \lambda_2 \mathcal{L}_{gan})}$ attains it with $(\varepsilon, \delta)$-edge-DP.*

With Corollary 1.2, we attain DPGGAN with the same $(\varepsilon, \delta)$-edge-DP protection of DPGVAE. For both DPGVAE and DPGGAN, the decoder/generator networks only get exposed to the noised and clipped gradients, representing the partial sensitive information within the training graphs. Hence, it prevents the inference of training graphs from both learned model parameters and generated graphs.

## 4 EXPERIMENTAL EVALUATIONS

We conduct two sets of experiments to evaluate the effectiveness of DPGGAN in *preserving global network structure* and *protecting individual link privacy*. All code and data are also in the submission.

**Experimental settings.** To provide a side-to-side comparison between the original networks and generated networks, we use two standard datasets of real-world networks, *i.e.*, DBLP, and IMDB. DBLP includes 72 networks of author nodes and co-author links, where the average numbers of nodes and links are 177.2 and 258; IMDB includes 1500 networks of actor/actress nodes and co-star links, with average node and link numbers 13 and 65.9.

To show that DPGGAN effectively captures global network structures, we compare it and DPG-VAE under different privacy budgets (controlled by $\varepsilon$ in Eq. (32)), regarding a suite of graph statistics commonly used to evaluate the performance of graph generation models, especially from a global perspective (Bojchevski et al., 2018; You et al., 2018b; Yang et al., 2019).[1] In particular, we train all models from scratch to convergence for $K$ times, where $K$ is the number of networks in the datasets. Each time, the trained model is used to generate one network, which is compared with the original network regarding the suite of graph statistics. Then we average the absolute differences between the generated networks and the original networks, ensuring that the positive and negative differences do not cancel out. The results are summarized in Table 1.

Beyond the single value statistics, we also compare the generated graph regarding degree distribution and motif counts. For degree distribution, we convert each graph into a 50-dim vector (all nodes with degree larger than 50 are binned together); for motif counts, we enumerate all 29 undirected motifs with 3-5 nodes and convert each graph into a 29-dim vector by motif matching. We compute the average cosine similarity between pairs of original graphs and generated graphs. Furthermore, we use the most widely studied graph-level downstream task, *i.e.*, graph classification, to evaluate the global utilities of generated graphs. In particular, we evaluate the accuracy of the state-of-the-art graph classification model, *i.e.*, GIN (Xu et al., 2019), with the default parameter setting and 4:1 training-testing ratio. The results are summarized in Table 2.

To facilitate a better understanding towards how the graph statistics reflect the global network structure captured by the models, we also provide results of two recent deep network generation methods, *i.e.*, NetGAN (Bojchevski et al., 2018) and GraphRNN (You et al., 2018b), with default parameter settings and no DP constraints at all. In this experiment, we expect to see the more effective *structure-preserving* models generate networks that are *more similar* to the original ones regarding various global and distributional graph properties and achieve *high graph classification accuracy*, thus maintaining high network data utility.

To show that DPGGAN effectively guarantees individual link privacy, we train all models for another $K$ times on each dataset. Instead of complete networks, we randomly sample 80% of the original networks' links to train the models. After training and generation, we use degree distribution to align the nodes in the generated networks with those in the original networks. Then we evaluate the standard AUC metric on the task of individual link prediction by comparing links predicted in the generated networks and links hidden during training in the original networks. In this experiment, we expect to see the more effective *privacy-protecting* models generate networks that are *less useful* for predicting individual links in the original networks, thus guaranteeing network data privacy.

All experiments are done with four GeForce GTX 1080 GPUs and a 12-core 2.2GHz CPU. The training time of DP-enforced models is often slightly shorter due to early stops when the privacy budget runs out, (*e.g.*, a typical train of GVAE, DPGVAE, and DPGGAN takes 60, 42 and 53 seconds on average on DBLP). The generation times of the three models are roughly the same (*e.g.*, 0.02 second on average on DBLP). As a direct comparison, NetGAN and GraphRNN take longer times under the same settings, especially for the generation (*e.g.*, 89, and 4.5 seconds for NetGAN to train and generate, and 75 and 2.4 seconds for GraphRNN, on DBLP). Although efficiency is not our primary concern, short runtimes (especially for generation) are favorable for efficient data share.

Due to space limitation, detailed settings of the neural architectures and hyper-parameters of our models are put into Appendix C.

---

[1]Statistics we use including LCC (size of the largest connected component), TC (triangle count), CPL (characteristic path length), GINI (gini index) and REDE (relative edge distribution entropy).

| Models | DBLP Networks | | | | | IMDB Networks | | | | |
|---|---|---|---|---|---|---|---|---|---|---|
| | LCC | TC | CPL | GINI | REDE | LCC | TC | CPL | GINI | REDE |
| Original | 107.5 | 59.90 | 3.6943 | 0.3248 | 0.9385 | 13.001 | 305.9 | 1.2275 | 0.1222 | 0.9894 |
| GVAE(no DP) | **7.51** | 66.93 | **0.1330** | **0.0213** | **0.0084** | 0.0145 | 25.83 | **0.0121** | 0.0030 | **0.0016** |
| NetGAN(no DP) | **9.66** | 39.87 | **0.1943** | **0.0105** | **0.0022** | 0.0083 | 27.54 | 0.0192 | 0.0042 | **0.0011** |
| GraphRNN(no DP) | **10.27** | 57.43 | 0.2043 | 0.0415 | **0.0052** | 0.0594 | 27.26 | 0.0214 | 0.0155 | 0.0094 |
| DPGVAE($\varepsilon$=10) | 21.96 | 175.29 | 0.2471 | 0.0339 | 0.0153 | 0.0147 | 43.63 | 0.0367 | 0.0036 | 0.0030 |
| DPGVAE($\varepsilon$=1) | 23.80 | 187.20 | 0.3059 | 0.0343 | 0.0156 | 0.0253 | 43.73 | 0.0373 | 0.0038 | 0.0031 |
| DPGVAE($\varepsilon$=0.1) | 26.07 | 215.13 | 0.3342 | 0.0344 | 0.0158 | 0.0320 | 44.12 | 0.0392 | 0.0042 | 0.0032 |
| DPGGAN($\varepsilon$=10) | 10.61 | **64.75** | **0.2035** | **0.0224** | 0.0093 | **0.0040** | **22.89** | 0.0164 | **0.0010** | **0.0017** |
| DPGGAN($\varepsilon$=1) | 12.38 | 70.97 | 0.2643 | 0.0353 | 0.0117 | **0.0053** | **23.81** | 0.0168 | 0.0029 | 0.0023 |
| DPGGAN($\varepsilon$=0.1) | 24.62 | 77.41 | 0.2713 | 0.0485 | 0.0191 | **0.0113** | **24.91** | **0.0168** | **0.0029** | 0.0025 |

Table 1: Performance evaluation over compared models regarding a suite of important graph structural statistics. The Original rows include the values of original networks, while the rest rows are the average absolute difference between generated networks by different models and the original networks. Therefore, *smaller values* indicate better capturing of global network structure and thus *better global data utility*. Bold font is used for values ranked top-3.

| Models | DBLP Networks | | | IMDB Networks | | |
|---|---|---|---|---|---|---|
| | Degree dist. | Motif ct. | GIN acc. | Degree dist. | Motif ct. | GIN acc. |
| GVAE(no DP) | **0.6171** | 0.4093 | 0.3029 | **0.5132** | **0.4129** | **0.4698** |
| NetGAN(no DP) | 0.5754 | **0.4109** | **0.3471** | 0.4921 | 0.3891 | 0.4350 |
| GraphRNN(no DP) | 0.5454 | 0.3672 | 0.3210 | 0.4635 | 0.3721 | 0.3875 |
| DPGVAE($\varepsilon$=1) | 0.5476 | 0.4038 | 0.3043 | 0.5081 | 0.4021 | 0.4625 |
| DPGGAN($\varepsilon$=1) | **0.6092** | **0.4150** | **0.3261** | **0.5486** | **0.4150** | **0.4725** |

Table 2: Performance evaluation regarding degree distribution, motif counts and GIN accuracy. *Larger values* for both cosine similarity and classification accuracy indicate *better graph utility*. Bold font is used for values ranked top-2.

**Preserving global structures.** In Table 1, our strictly DP-constrained models constantly yield highly competitive and even better results compared with the strongest DP-free baselines regarding global network structural similarity between generated and original networks on both datasets, clearly showing the effectiveness of our models on global network structure preservation. As we gradually increase the privacy budget $\varepsilon$, our two models (especially DPGGAN) apparently perform better, showing the effectiveness of our privacy constraints and a clear trade-off between privacy and utility. Furthermore, as in Table 2, the graphs generated by DPGGAN are competitively similar to the original graphs regarding both degree distributions and motif counts, while achieving satisfactory graph classification accuracy. The improvements of DPGGAN all passed t-tests with p-value 0.01, which corroborates our novel design of the structure-oriented graph generation framework.

**Protecting individual links.** For both datasets, links predicted on the networks generated by DPG-GAN are much *less accurate* than those predicted on the original networks (26%-35% and 15%-20% AUC drops on DBLP and IMDB, respectively) as well as the networks generated by all baselines. This means even if the attackers identify nodes in the generated (released) networks of DPGGAN, they cannot leverage the information there to accurately infer the existence or absence of links between particular pairs of nodes on the original networks. This directly corroborates our claim that DPGGAN is effective in protecting individual link privacy.

Due to space limit, more details and discussions regarding the experimental results are put into Appendix D. In Appendix E, we also provide graph visualizations for qualitative visual inspections.

## 5 CONCLUSION

Due to the recent development of deep graph generation models, synthetic networks are generated and released for granted, without the concern about possible privacy leakage over the original networks used for model training. In this work, for the first time, we pay attention to the task of secure network release and formulate its goals as *preserving global network structure* while *protecting individual link privacy*. Subsequently, we adopt the well-studied DP framework and develop DPGGAN, which protects individual link privacy by enforcing edge-DP on the graph generation model while preserving global network structure with a structure-oriented graph discriminator. Comprehensive experiments show that DPGGAN is advantageous in generating networks that are globally similar to the original ones (thus effectively maintaining network data utility), and at the same time, useless for predicting individual links in the original network (thus rigorously protecting network data privacy).

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

# A  APPENDIX: PROOFS FOR THEOREM 1

In this appendix, we provide proofs for Theorem 1, and derive Corollary 1.1 and Corollary 1.2. Theorem 1 indicates the link privacy protection achieved through updating model's parameters with clipped and noised gradient (latter referred to as DP learning) for link reconstruction based graph generation models. Corollary 1.1 and Corollary 1.2 derived from Theorem 1 support us to guarantee $(\epsilon, \delta)$-edge-DP for DPGVAE and DPGGAN with DP learning in Theorem 1.

The proof for Theorem 1 is divided into three steps. We first briefly introduce the definition of the moment accountant privacy analysis and respective properties in Section A.1, for it being the fundamentals of our proof. Note that in (Abadi et al., 2016), DPSGD is originally designed for classical machine learning tasks, such as image classification. Therefore, in Section A.2, we leverage moment accountant to conduct the extended privacy analysis of DPSGD for general types of data and loss functions. Then in Section A.3, we apply the conclusion from Section A.2 on graph data and the link reconstruction loss function to derive the theoretical analysis over edge-DP achieved by link reconstruction based graph generation models and finish our proof for Theorem 1. Following the conclusion in Theorem 1, we tune gradient representations to certain gradient functions leveraged in training DPGVAE decoder and DPGGAN generator to derive Corollary 1.1 and Corollary 1.2, as the theoretical support for the $(\varepsilon, \delta)$-edge-DP held by respective models.

## A.1  MOMENT ACCOUNTANT

Our proof for Theorem 1 is mainly based on moment accountant (Abadi et al., 2016). The definition of moment accountant and the properties leveraged in our proof are listed below.

**Definition 3.** *Let $\mathcal{M} : \mathcal{D} \to \mathcal{R}$ be a randomized mechanism and $d, d'$ a pair of adjacent databases. Let $aux$ denote an auxiliary input. For an outcome $o \in \mathcal{R}$, the privacy loss at $o$ is defined as:*

$$c\left(o; \mathcal{M}, aux, d, d'\right) \triangleq \log \frac{\Pr[\mathcal{M}(aux, d) = o]}{\Pr\left[\mathcal{M}\left(aux, d'\right) = o\right]} \tag{7}$$

*The privacy loss random variable $C\left(\mathcal{M}, aux, d, d'\right)$ is defined as $c\left(\mathcal{M}(d); \mathcal{M}, aux, d, d'\right)$, i.e.the random variable defined by evaluating the privacy loss at an outcome sampled from $\mathcal{M}(d)$.*

**Definition 4.** *Let $\mathcal{M} : \mathcal{D} \to \mathcal{R}$ be a randomized mechanism and $d, d'$ a pair of adjacent databases. Let $aux$ denote an auxiliary input. The moments accountant is defined as:*

$$\alpha_{\mathcal{M}}(\lambda) \triangleq \max_{aux, d, d'} \alpha_{\mathcal{M}}\left(\lambda; aux, d, d'\right) \tag{8}$$

*where $\alpha_{\mathcal{M}}\left(\lambda; aux, d, d'\right) \triangleq \log \mathbb{E}\left[\exp\left(\lambda C\left(\mathcal{M}, aux, d, d'\right)\right)\right]$ is the moment generating function of the privacy loss random variable.*

The following properties of the moments accountant are proved in (Abadi et al., 2016).

**Property 4.1.** *[Composability] Suppose that a mechanism $\mathcal{M}$ consists of a sequence of adaptive mechanisms $\mathcal{M}_1, \ldots, \mathcal{M}_k$ where $\mathcal{M}_i : \prod_{j=1}^{i-1} \mathcal{R}_j \times \mathcal{D} \to \mathcal{R}_i$. Then, for any output sequence $o_1, \ldots, o_{k-1}$ and any $\lambda$, we have*

$$\alpha_{\mathcal{M}}\left(\lambda; d, d'\right) = \sum_{i=1}^{k} \alpha_{\mathcal{M}_i}\left(\lambda; o_1, \ldots, o_{i-1}, d, d'\right) \tag{9}$$

*where $\alpha_{\mathcal{M}}$ is conditioned on $\mathcal{M}_i$'s output being $o_i$ for $i < k$.*

**Property 4.2.** *[Tail bound] For any $\varepsilon > 0$, the mechanism $\mathcal{M}$ is $(\varepsilon, \delta)$-DP for*

$$\delta = \min_{\lambda} \exp\left(\alpha_{\mathcal{M}}(\lambda) - \lambda\varepsilon\right) \tag{10}$$

## A.2  THE GENERALIZED PRIVACY ANALYSIS OF DPSGD

To achieve $(\epsilon, \delta)$-edge-DP for graph data, we exploit DPSGD (Abadi et al., 2016) with necessary adaptions according to the special nature of graph data compared to other types of data (*e.g.*, images), for which DPSGD was originally designed. The original DPSGD only provides DP proof for gradient function $f$ clipped by $C$ with its $\ell_2$-norm sensitivity as $\Delta_2 f = 1 \cdot C = C$. For classical tasks of machine learning like image classification, $\Delta_2 f = C$ is obvious. However, in a more complex task like graph learning, a minor change in the training dataset can probably induce a different gap according to the chosen measurement. To explore the potential of DPSGD with customized machine learning tasks, we further prove the privacy performance of DPSGD with a gradient function $f$ with $\ell_2$-norm sensitivity $\Delta_2 f = s$.

Therefore, to prepare for the proof for Theorem 1, we first leverage moments accountant to derive the upper bound of privacy loss for a Gaussian Mechanism as below.

**Lemma 1.** *Suppose that $f : D \to \mathbb{R}^p$ with $\|f(\cdot)\|_2 \le s$. Let $\sigma \ge s$ and let $J$ be a sample from $[n]$ where each $i \in [n]$ is chosen independently with probability $q < \frac{s}{16\sigma}$. Then for any positive integer $\lambda \le \frac{\sigma^2}{s^2} \ln \frac{s}{q\sigma}$, the Gaussian Mechanism $\mathcal{M}(d) = \sum_{i \in J} f(d_i) + \mathcal{N}(0, \sigma^2 \mathbf{I})$ satisfies*

$$\alpha_\mathcal{M}(\lambda) \le \frac{s^2 q^2 \lambda(\lambda+1)}{(1-q)\sigma^2} + O\left(s^3 q^3 \lambda^3 / \sigma^3\right) \tag{11}$$

*Proof.* Fix $d'$ and let $d = d' \cup \{d_n\}$. Without loss of generality, we assume $f(d_n) = s \cdot \mathbf{e}_1$ and $\sum_{i \in J \setminus [n]} f(d_i) = \mathbf{0}$. Thus $\mathcal{M}(d)$ and $\mathcal{M}(d')$ are distributed identically except for the first coordinate and hence we have a one-dimensional problem. Let $\mu_0$ denote the pdf of $\mathcal{N}(0, \sigma^2)$ and let $\mu_s$ denote the pdf of $\mathcal{N}(s, \sigma^2)$. We have

$$\begin{aligned}\mathcal{M}(d') &\sim \mu_0, \\ \mathcal{M}(d) &\sim \mu \triangleq (1-q)\mu_0 + q\mu_s.\end{aligned} \tag{12}$$

We want to show that

$$\begin{aligned}&\mathbb{E}_{z \sim \mu}\left[(\mu(z)/\mu_0(z))^\lambda\right] \le \alpha, \\ \text{and } &\mathbb{E}_{z \sim \mu_0}\left[(\mu_0(z)/\mu(z))^\lambda\right] \le \alpha,\end{aligned} \tag{13}$$

where $\alpha$ is a value to be determined. We will use the same method as in (Abadi et al., 2016) to prove both bounds. Assume we have two distributions $\nu_0$ and $\nu_s$, and we wish to bound

$$\mathbb{E}_{z \sim \nu_0}\left[(\nu_0(z)/\nu_s(z))^\lambda\right] = \mathbb{E}_{z \sim \nu_s}\left[(\nu_0(z)/\nu_s(z))^{\lambda+1}\right]. \tag{14}$$

Leveraging binomial expansion, we obtain

$$\begin{aligned}&\mathbb{E}_{z \sim \nu_s}\left[(\nu_0(z)/\nu_s(z))^{\lambda+1}\right] \\ =&\mathbb{E}_{z \sim \nu_s}\left[(1 + (\nu_0(z) - \nu_s(z))/\nu_s(z))^{\lambda+1}\right] \\ =&\mathbb{E}_{z \sim \nu_s}\left[(1 + (\nu_0(z) - \nu_s(z))/\nu_s(z))^{\lambda+1}\right] \\ =&\sum_{t=0}^{\lambda+1} \binom{\lambda+1}{t} \mathbb{E}_{z \sim \nu_s}\left[((\nu_0(z) - \nu_s(z))/\nu_s(z))^t\right].\end{aligned} \tag{15}$$

The first term in Eq. (15) is 1, and the second term is

$$\begin{aligned}(\lambda+1)\mathbb{E}_{z \sim \nu_s}\left[\frac{\nu_0(z) - \nu_s(z)}{\nu_s(z)}\right] &= \int_{-\infty}^{+\infty} \nu_s(z)\frac{\nu_0(z) - \nu_s(z)}{\nu_s(z)}\mathrm{d}z \\ &= (\lambda+1)\int_{-\infty}^{+\infty} \nu_0(z)\mathrm{d}z - \int_{-\infty}^{+\infty} \nu_s(z)\mathrm{d}z \\ &= (\lambda+1)(1-1) = 0.\end{aligned} \tag{16}$$

Regarding conditions stated in the lemma, for both cases, where $\nu_0 = \mu, \nu_1 = \mu_0$ and $\nu_0 = \mu_0, \nu_1 = \mu$, the third term is bounded by $q^2\lambda(\lambda+1)/(1-q)\sigma^2$ and this bound dominates the sum of the remaining terms. We provide the proof for the case of $(\nu_0 = \mu_0, \nu_s = \mu)$, and the proof of the other case is similar.

To upper bound the third term in 15, we note that $\mu(z) \geq (1-q)\mu_0(z)$, and write

$$
\begin{aligned}
\mathbb{E}_{z\sim\mu}&\left[\left(\frac{\mu_0(z)-\mu(z)}{\mu(z)}\right)^2\right]\\
&=q^2\mathbb{E}_{z\sim\mu}\left[\left(\frac{\mu_0(z)-\mu_s(z)}{\mu(z)}\right)^2\right]\\
&=q^2\int_{-\infty}^{+\infty}\frac{(\mu_0(z)-\mu_s(z))^2}{\mu(z)}\mathrm{d}z\\
&\leq\frac{q^2}{1-q}\int_{-\infty}^{+\infty}\frac{(\mu_0(z)-\mu_s(z))^2}{\mu_0(z)}\mathrm{d}z\quad=\frac{q^2}{1-q}\mathbb{E}_{z\sim\mu_0}\left[\left(\frac{\mu_0(z)-\mu_s(z)}{\mu_0(z)}\right)^2\right].
\end{aligned}
\tag{17}
$$

Recalling the definition of $\mu_0$ and the normal distribution, we have

$$
\begin{aligned}
\mathbb{E}_{z\sim\mu_0}\left[\left(\frac{\mu_0(z)-\mu_1(z)}{\mu_0(z)}\right)^2\right]&=\mathbb{E}_{z\sim\mu_0}\left[\left(1-\exp\left(\frac{2sz-s^2}{2\sigma^2}\right)\right)^2\right]\\
&=1-2\mathbb{E}_{z\sim\mu_0}\left[\exp\left(\frac{2sz-s^2}{2\sigma^2}\right)\right]+\mathbb{E}_{z\sim\mu_0}\left[\exp\left(\frac{4sz-2s^2}{2\sigma^2}\right)\right].
\end{aligned}
\tag{18}
$$

For the second term in Eq. (18) $\mathbb{E}_{z\sim\mu_0}\left[\exp\left(\frac{2sz-s^2}{2\sigma^2}\right)\right]$, we have

$$
\mathbb{E}_{z\sim\mu_0}\left[\exp\left(\frac{2sz-s^2}{2\sigma^2}\right)\right]=\int_{-\infty}^{+\infty}\frac{1}{\sigma\sqrt{2\pi}}\exp\left(\frac{-(z-s)^2}{2\sigma^2}\right)\mathrm{d}z=1.
\tag{19}
$$

For the third term in Eq. (18), we have

$$
\begin{aligned}
\mathbb{E}_{z\sim\mu_0}\left[\exp\left(\frac{4sz-2s^2}{2\sigma^2}\right)\right]&=\exp\left(\frac{s^2}{\sigma^2}\right)\int_{-\infty}^{+\infty}\frac{1}{\sigma\sqrt{2\pi}}\exp\left(\frac{-(z-2s)^2}{2\sigma^2}\right)\mathrm{d}z\\
&=\exp\left(\frac{s^2}{\sigma^2}\right).
\end{aligned}
\tag{20}
$$

Thus, for Eq. (18), we have

$$
\mathbb{E}_{z\sim\mu_0}\left[\left(\frac{\mu_0(z)-\mu_1(z)}{\mu_0(z)}\right)^2\right]=\exp\left(s^2/\sigma^2\right)-1.
\tag{21}
$$

Hence, the third term in the binomial expansion of Eq. (15) is

$$
\binom{1+\lambda}{2}\mathbb{E}_{z\in\mu}\left[\left(\frac{\mu_0(z)-\mu(z)}{\mu(z)}\right)^2\right]\leq\frac{\lambda(\lambda+1)q^2}{2(1-q)}\left(\exp(\frac{s^2}{\sigma^2})-1\right)
\tag{22}
$$

For $\sigma\geq s$, it is easy to get $\exp(\frac{s^2}{\sigma^2})-1\leq\frac{2s^2}{\sigma^2}$. Therefore, we retrieve that

$$
\binom{1+\lambda}{2}\mathbb{E}_{z\in\mu}\left[\left(\frac{\mu_0(z)-\mu(z)}{\mu(z)}\right)^2\right]\leq\frac{\lambda(\lambda+1)q^2s^2}{(1-q)\sigma^2}.
\tag{23}
$$

By standard calculus, we get $|\mu_0(z)-\mu_s(z)|=\left|\int_{z-s}^z\mu_0'(z)\mathrm{d}z\right|$. Note that $\mu_0'(z)$ is monotonically decreasing in $(-\infty,+\infty)$. Thus, to bound the remaining terms, we derive

$$
\begin{aligned}
\forall z\leq 0:&|\mu_0(z)-\mu_s(z)|\leq-s(z-s)\mu_s(z)/\sigma^2\\
\forall z\geq s:&|\mu_0(z)-\mu_s(z)|\leq zs\mu_0(z)/\sigma^2\\
\forall 0\leq z\leq s:&|\mu_0(z)-\mu_s(z)|\leq\mu_0(z)\left(\exp\left(s^2/2\sigma^2\right)-1\right)\\
&\qquad\qquad\qquad\leq s^2\mu_0(z)/\sigma^2.
\end{aligned}
\tag{24}
$$

We can then write

$$\mathbb{E}_{z \sim \mu}\left[\left(\frac{\mu_0(z) - \mu(z)}{\mu(z)}\right)^t\right]$$

$$\leq \int_{-\infty}^0 \mu(z)\left|\left(\frac{\mu_0(z) - \mu(z)}{\mu(z)}\right)^t\right|\mathrm{d}z$$

$$+ \int_0^s \mu(z)\left|\left(\frac{\mu_0(z) - \mu(z)}{\mu(z)}\right)^t\right|\mathrm{d}z \qquad (25)$$

$$+ \int_s^{+\infty} \mu(z)\left|\left(\frac{\mu_0(z) - \mu(z)}{\mu(z)}\right)^t\right|\mathrm{d}z.$$

We consider these terms individually. We repeatedly make use of three observations: (1) $\mu_0 - \mu = q(\mu_0 - \mu_s)$,(2)$\mu \geq (1-q)\mu_0$, (3)$\mu \geq q\mu_s$, and (4) $\mathbb{E}_{\mu_0}\left[|z|^t\right] \leq \sigma^t(t-1)!!$. The first term can then be bounded by

$$\frac{q^t}{(1-q)^{t-1}\sigma^{2t}} \int_{-\infty}^0 \mu_0(z)|z-1|^t\mathrm{d}z$$

$$\leq \int_{-\infty}^0 q\mu_s\left|\left(\frac{\mu_0 - \mu_s}{\mu_s}\right)^t\right|\mathrm{d}z$$

$$\leq \frac{qs^t}{\sigma^{2t}} \int_{-\infty}^0 \mu_s\left|(z-s)^t\right|\mathrm{d}z \qquad (26)$$

$$\leq \frac{qs^t(t-1)!!}{2\sigma^t}.$$

Then the second term is at most

$$\frac{q^t}{(1-q)^t} \int_0^s \mu(z)\left|\left(\frac{\mu_0(z) - \mu_1(z)}{\mu_0(z)}\right)^t\right|\mathrm{d}z \leq \frac{q^t}{(1-q)^t} \int_0^s \mu(z)\left|(s^2/\sigma^2)^t\right|\mathrm{d}z$$

$$\leq \frac{q^t s^{2t}}{(1-q)^t\sigma^{2t}}. \qquad (27)$$

Similarly, the third term is at most

$$\frac{q^t s^t}{(1-q)^{t-1}\sigma^{2t}} \int_s^{+\infty} \mu_0(z)\left|z^t\right|\mathrm{d}z \leq \frac{q^t s^t(t-1)!!}{(1-q)^{t-1}\sigma^t}. \qquad (28)$$

Under the assumptions on $q$, $\sigma$, and $\lambda$, it is easy to check that the three terms, and their sum, drop off geometrically fast in $t$ for $t > 3$. Hence the binomial expansion (5) is dominated by the $t = 3$ term, which is $O\left(s^3 q^3 \lambda^3/\sigma^3\right)$. Therefore, the lemma is proved. □

With Lemma 1, we retrieve the upper bound of privacy loss of the Gaussian Mechanism. Hence, based on Lemma 1 and Property 4.1, we provide the generalized privacy analysis of DPSGD with different learning tasks, which iteratively performs multiple times of the Gaussian Mechanism.

**Lemma 2.** *Suppose that $f : D \to \mathbb{R}^p$ with $\|f(\cdot)\|_2 \leq s$. Let $J$ be a sample from $[N]$ that each $i \in [N]$ is chosen independently in probability $q = |J|/N$, given the number of steps $T$, for any $c_0 \in (0, 1)$, there exist explicit constants $c_1$ and $c_2$ that with any $\varepsilon < c_1 q^2 T$, iteratively computing $T$ times of $\mathcal{M}(d)$ in Lemma 1 attains it with $(\varepsilon, \delta)$-DP for any $\delta > 0$ if we choose*

$$\sigma \geq c_2 \frac{qs\sqrt{T\log(1/\delta)}}{\varepsilon}, \qquad (29)$$

*where $c_1 \geq \frac{1}{c_0}\log\frac{s}{q\sigma}$ and $c_2 \leq \frac{1}{\sqrt{c_0(1-c_0)}}$ for any $c_0 \in (0, 1)$.*

*Proof.* Assume for now that $\sigma, \lambda$ satisfy the conditions in Lemma 1. After $T$ times of iteration, with Property 4.1 we derive that $\alpha(\lambda) \leq Tq^2 s^2 \lambda^2/\sigma^2$. In order to to guarantee the whole training process to be $(\varepsilon, \delta)$ -DP, combining $\alpha(\lambda)$ with Property 4.2, for any $c_0 \in (0, 1)$, we choose

$$Tq^2 s^2 \lambda^2/\sigma^2 = c_0 \lambda \varepsilon,$$
$$\exp((c_0 - 1)\lambda\varepsilon) \leq \delta. \qquad (30)$$

Plugging the condition $\lambda \leq \frac{\sigma^2}{s^2} \log \frac{s}{q\sigma}$ into Eq. (30), we derive the bound for $\varepsilon$ as $\varepsilon < \frac{1}{c_0} \log \frac{s}{q\sigma} q^2 T$ to accomplish $(\varepsilon, \delta)$-DP by setting

$$\sigma = \frac{1}{\sqrt{c_0(1-c_0)}} \cdot \frac{qs\sqrt{T\log(1/\delta)}}{\varepsilon},$$  (31)

where $c_0 \in (0, 1)$. □

## A.3 PRIVACY ANALYSIS FOR THE LINK RECONSTRUCTION BASED GRAPH GENERATION MODELS WITH DPSGD

In this section, we conduct the theoretical privacy analysis for link reconstruction based graph generation model based on Lemma 2 and obtain the conclusion of Theorem 1.

**Theorem 1.** *In training a link reconstruction based graph generation model on a graph with $N$ nodes with batch size as $B$, given the sampling probability $q = B/N$, and the number of steps $T$, there exist explicit constants $c_1$ and $c_2$ that for any $\varepsilon < c_1 q^2 T$, iteratively updating the model $T$ times with $\tilde{g}_{\theta, \mathcal{L}}$ attains it with $(\varepsilon, \delta)$-edge-DP for any $\delta > 0$ if we choose*

$$\sigma \geq c_2 \frac{q\sqrt{T\log(1/\delta)}}{\varepsilon},$$

*where $c_1 \geq \frac{1}{c_0} \log \frac{1}{q\sigma}$, $c_2 \leq 1/\sqrt{c_0(1-c_0)}$ for any $c_0 \in (0, 1)$.*

*Proof.* Recall the expression of $\tilde{g}_{\theta, \mathcal{L}}$ as in Eq. (5)

$$\tilde{g}_{\theta, \mathcal{L}} = \frac{1}{N} \left( \sum_{i=1}^{N} \left( \nabla_{v_i, \theta} \mathcal{L} / \max(1, \frac{\|\nabla_{v_i, \theta} \mathcal{L}\|_2}{C}) \right) + \mathcal{N}(0, \sigma^2 C^2 \mathbf{I}) \right),$$

where $\mathcal{L}$ is the loss function for a link reconstruction based graph generation model, $C$ is the clipping hyper-parameter for the model's original gradient to bound the influence of each node, and $\sigma$ is the noise scale hyper-parameter. According to Definition 1, $\tilde{g}_{\theta, \mathcal{L}}$ is a Gaussian mechanism. Therefore, we first analyze the $\ell_2$-norm sensitivity of the clipped gradient function $\tilde{g}_{\theta, \mathcal{L}}$, and then plug the sensitivity value to Lemma 2 and conclude the privacy cost of training DPGVAE, thus finishing the proof for Thereom 1.

Following the graph reconstruction procedure in (Simonovsky & Komodakis, 2018), a single value in the adjacency matrix is sufficient to represent one edge in the respective graph for both directed and undirected graphs. Therefore, referring to Definition 2, though changing an edge in the graph affects 2 nodes for node classification tasks, for a structural inference task, *i.e.*, graph reconstruction, as our work targeting at, adding or removing an edge only results to at most 1 record difference. Together with $\nabla_{v_i, \mathbf{f}} \mathcal{L}$ being clipped as its $\ell_2$-norm no more than $C$, we obtain the sensitivity of $\sum_{i=1}^{N} \nabla_{v_i, \mathbf{f}} \mathcal{L} / \max(1, \frac{\|\nabla_{v_i, \mathbf{f}} \mathcal{L}\|_2}{C})$ as $C$.

With plugging in the clipped $\nabla_{v_i, \mathbf{f}} \mathcal{L}$'s sensitivity ($s = C$) into Lemma 2, we derive Theorem 1. We prove that, given the sampling probability $q = B/N$ and the number of steps $T$, with explicit constants $c_1 \geq \frac{1}{c_0} \log \frac{1}{q\sigma}$ and $c_2 \leq \frac{1}{\sqrt{c_0(1-c_0)}}$, where $c_0 \in (0, 1)$, through iteratively updating model $T$ times with Eq. (5), the outcome generation model achieves $(\varepsilon, \delta)$-edge-DP for any $\varepsilon < c_1 q^2 T$, and $\delta > 0$ when we choose

$$\sigma \geq c_2 \frac{q\sqrt{T\log(1/\delta)}}{\varepsilon}.$$  (32)

□

Recall the training process for the decoder in DPGVAE and the generator in DPGGAN in Section 3. $\mathcal{L}$ in $\tilde{g}_{\theta, \mathcal{L}}$ is substituted with $\mathcal{L}_{rec}$ and $\mathcal{L}_{rec} + \lambda_2 \mathcal{L}_{gan}$, respectively. For both $\mathcal{L}_{rec}$ and $\mathcal{L}_{rec} + \lambda_2 \mathcal{L}_{gan}$, their gradients are clipped with $C$ and adding Gaussian noises during the training process. Based on Theorem 1, we derive Corollary 1.1 and 1.2 for the decoder in DPGVAE and the generator in DPGGAN respectively as below.

**Corollary 1.1** (DPGVAE edge-DP). *Under the same conditions in Theorem 1, iteratively updating the decoder in DPGVAE $T$ times with $\tilde{g}_{\theta, \mathcal{L}_{rec}}$ attains it with $(\varepsilon, \delta)$-edge-DP.*

**Corollary 1.2** (DPGGAN edge-DP). *Under the same conditions in Theorem 1, iteratively updating the generator in DPGGAN $T$ times with $\tilde{g}_{\theta, (\mathcal{L}_{rec} - \lambda_2 \mathcal{L}_{gan})}$ attains it with $(\varepsilon, \delta)$-edge-DP.*

With Corollary 1.1 and 1.2, under specified conditions, the public model (either the decoder in DPGVAE or the generator in DPGGAN) is guaranteed with $(\varepsilon, \delta)$-edge-DP by the DP training process. For both DPGVAE decoder and DPGGAN generator updated with noised and clipped representations of the sensitive training graph,

they only record noised and partial sensitive information. DPGVAE decoder and DPGGAN generator's link reconstruction procedures, reflecting its training information, only allude to the desensitize information rather than the true sensitive training information. Thus, DPGVAE decoder and DPGGAN generator not only prevent privacy leakage from their inner parameters with DP learning but also preserve the raw private training graphs from being accurately inferred through the respective outputs.

## B  APPENDIX: DETAILED TRAINING ALGORITHM

The overall architecture of DPGGAN is shown in Figure 2 in the main content. In Figure 2, the original graph is fed into the GCN-based encoder network $\mathbf{g}$ to compute node embeddings, which is then compared with the prior distribution $\mathcal{N}(\mathbf{0}, \mathbf{I})$ and fed into the FNN-based decoder/generator network $\mathbf{f}$ to produce the reconstructed graph and generated graph. After going through the GCN-based discriminator network part 1 ($\mathbf{g}'$), a reconstruction loss is computed between the reconstructed graph and the original graph, and a discrimination loss is computed for the generated graph and original graph after the FNN-based discriminator network part 2 ($\mathbf{f}'$).

---

**Algorithm 1** DPGGAN

**Input** : Graph data $\mathbf{G}(\mathbf{A}, \mathbf{X})$, clipping parameter $C$, decay ratio $\gamma$, privacy budget $\varepsilon$, noise scale $\sigma$, total number of nodes $N$, batch size $B = qN$, learning rate $\eta$, maximum number of training epochs $T$, loss weighing parameters $\lambda_1$ and $\lambda_2$

**Output**: Differentially private decoder $\mathbf{f}$.

1 Initialize weights randomly for $\mathbf{g}_\mu, \mathbf{g}_\sigma, \mathbf{f}, \mathbf{g}'$ and $\mathbf{f}'$.
2 **for** *epoch $t = 0$ to $T$* **do**
3  **for** *iteration $i = 0$ to $N/B$* **do**
4    Sample a subgraph $\mathbf{G}_{sub}(\mathbf{A}_{sub}, \mathbf{X}_{sub})$ of size $B$
5    Mean vector: $\mu \leftarrow \mathbf{g}_\mu(\mathbf{X}_{sub}, \mathbf{A}_{sub})$
6    Standard deviation vector: $\sigma \leftarrow \mathbf{g}_\sigma(\mathbf{X}_{sub}, \mathbf{A}_{sub})$
7    Update $q(\mathbf{Z}|\mathbf{X}, \mathbf{A}) \leftarrow \prod_{i=1}^{N} \mathcal{N}(\mathbf{z}_i|\mu_i, \mathrm{diag}(\sigma_i^2))$
8    Sample $\mathbf{z}_i, \mathbf{z}_j \sim q(\mathbf{Z}|\mathbf{X}, \mathbf{A})$
9    Reconstruct adjacent matrix $\mathbf{A}' \leftarrow \sigma(\mathbf{f}(\mathbf{z}_i)^T, \mathbf{f}(\mathbf{z}_j))$
10   $\mathcal{L}_{\text{prior}} = D_{\text{KL}}(q(\mathbf{Z}|\mathbf{X}, \mathbf{A})\|p(\mathbf{Z}))$
11   $\mathcal{L}_{gan} = \log(\mathcal{D}(\mathbf{A})) + \log(1 - \mathcal{D}(\mathbf{A}'))$
12   **for** *node $x_i \in \mathbf{G}_{sub}$* **do**
13     Compute $g_\theta(x_i) \leftarrow \partial(\mathcal{L}_{rec} - \lambda_2 \mathcal{L}_{gan})/\partial x_i$
14   **end**
15   Clip gradient: $\bar{g}_\theta(x_i) \leftarrow g_\theta(x_i)/max(1, \frac{\|g_\theta(x_i)\|_2}{C})$
16   Perturb gradient $\tilde{g}_\theta \leftarrow \frac{1}{B}(\sum_i \bar{g}_\theta(x_i) + \mathbf{N}(\mathbf{0}, \sigma^2 C^2 \mathbf{I}))$
17   Average gradient: $g_\theta \leftarrow \frac{1}{B} \sum_i g_\theta(x_i)$
18   Update $\mathbf{f} \xleftarrow{+} \eta \cdot \tilde{g}_\theta$; $\mathbf{f}', \mathbf{g}' \xleftarrow{+} \eta \cdot \nabla_{\mathbf{g}' \cdot \mathbf{f}'} \lambda_2 \mathcal{L}_{gan}$;   //apply DP learning to the generator
19   Update $\mathbf{g}_\mu, \mathbf{g}_\sigma \xleftarrow{+} \eta \cdot \nabla_{\mathbf{g}}(\mathcal{L}_{rec} + \lambda_1 \mathcal{L}_{prior})$
20  **end**
21  Update $\mathbf{C} = \mathbf{C} * \gamma$
22 **end**

---

Here, we give more details towards its training procedures in Algorithm 1. For a better description, we shorten $g_{\theta,(\mathcal{L}_{rec}-\lambda_2\mathcal{L}_{gan})}$ ($\tilde{g}_{\theta,(\mathcal{L}_{rec}-\lambda_2\mathcal{L}_{gan})}$) as $g_\theta$ ($\tilde{g}_\theta$). In the algorithm, for proper gradient distortion, we devise gradient clipping in Line 15 and noise injection in Line 16, which is only applied to the generator network $\mathbf{f}$ in Line 18. In Line 21, we gradually reduce the clipping hyper-parameter $C$ as the volume of gradients decreases along training.

In the experimental analysis, existing works often fix $\delta$ as a dataset-specific value like $10^{-5}$, and then analyze the performance of models based on fixed privacy budget $\varepsilon$ (Abadi et al., 2016) or fixed noise scale $\sigma$ (Papernot et al., 2018). According to Theorem 1, our experiments are conducted with fixing the noise scale $\sigma = 2q\sqrt{T \log(1/\delta)}/\varepsilon$, where $\varepsilon < 2 \log \frac{1}{q\sigma} q^2 T$. In this work, other than the noise scale $\sigma$, to control the variance, we also fix $\delta$, the sampling ratio $q$, for better analysis of the model's privacy loss. Note that we vary the number of training iterations (query number) $T$ to study the relation between the model's performance and the privacy budget.

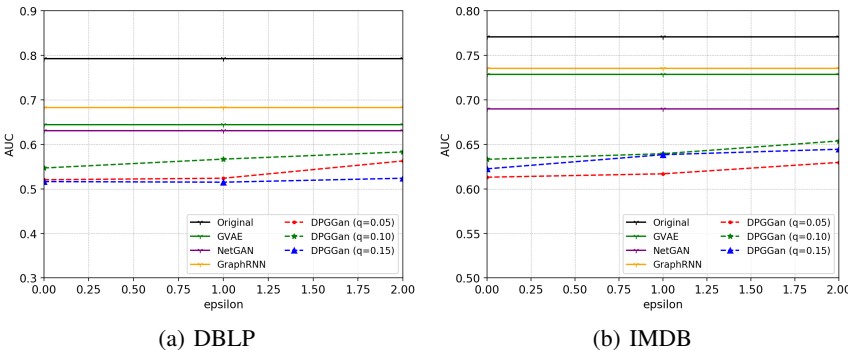

(a) DBLP                                     (b) IMDB

Figure 3: Accuracy of links predicted based on networks generated by DPGGAN with varying hyper-parameters and evaluated on the original networks. *Lower AUC* means the information in the generated networks is less useful in revealing the true existence or absence of links in the original networks, thus *better individual data privacy*.

## C   APPENDIX: MORE DETAILS OF EXPERIMENTAL SETTINGS

For GVAE and our models, we use two-layer GCNs with sizes $32 \to 16$ for both $\mathbf{g}_\mu$ and $\mathbf{g}_\sigma$ of the encoder network, where the first layer is shared, and we use two-layer FNNs with sizes $16 \to 32$ for $\mathbf{f}$ of the decoder (generator) network. For DPGGAN, we use another two-layer GCN with the same sizes for $\mathbf{g}'$ and a three-layer FNN with sizes $16 \to 32 \to 1$ for $\mathbf{f}'$. For DP-related hyper-parameters, we follow existing works (Abadi et al., 2016; Shokri & Shmatikov, 2015) to fix $\delta$ to $10^{-5}$, $\sigma$ to 5, and $q$ to 0.01 (which determines the batch size $B$ as $B = qN$ with $N$ as the graph size). Then we vary $\varepsilon$ from 0.1 to 10 to see how much graph-level utilities are preserved under different privacy budgets. According to Eq. (32), we terminate the training of DPGGAN at $T$ iterations when $\varepsilon$ is depleted. Other than the essential parameters in Eq. (32), we empirically set the clipping parameter $C$ to 5, decay ratio $\gamma$ to 0.99, learning rate $\eta$ to $10^{-3}$, and the loss weighing parameters $\lambda_1$ and $\lambda_2$ both to 0.1. We do not observe the model to be very sensitive to the setting of these non-essential parameters.

## D   APPENDIX: MORE DETAILS OF EXPERIMENTAL RESULTS

In this work, we define the goals of secure network release as *preserving global network structure* while *protecting individual link privacy*. In the main content, we have presented experimental results to support the effectiveness of DPGGAN in both perspectives. That is, for global network structure preservation, we show that the generated graphs of DPGGAN are competitively similar to the original graphs in comparison with the DP-free state-of-the-art graph generative models regarding a suite of commonly concerned global graph statistics, and for individual link privacy protection, we show that the links predicted in the generated graphs of DPGGAN are useless (with low accuracy) when evaluated in the original graphs.

The suite of statistics measures the global network structure from different perspectives. As can be inferred from TC, CPL, and GINI, the IMDB networks are in general smaller, tighter, and likely more structurally complex than the DBLP networks, which favors link generation models (*e.g.*, GVAE) over sequence generation models (*e.g.*, NetGAN, and GraphRNN). Consequently, DPGGAN also performs better on the IMDB networks, indicating its advantages in modeling complex link structures as a whole.

In addition to the graph statistics, we further demonstrate the data utility of networks generated by DPGGAN with graph classification, which is the most widely studied graph-level downstream task. We deem this task important towards evaluating network data utility, especially under our consideration of global network structure preservation, because correct graph classification requires the generated graphs to share essential structural properties with the original graphs. As we can see from Table 2 in the main paper, the data utilities evaluated with graph classification are consistent with those evaluated with global graph statistics, as shown in Table 1 in the main paper. Our two DP-constrained models yield highly competitive performance compared with the DP-free state-of-the-art graph generative models.

As for privacy protection, we conduct more detailed inspections of the performance of individual link prediction. In particular, we vary two of the major hyper-parameters, *i.e.*, the privacy budget $\varepsilon$, and sampling ratio $q$. Consistently with the results in Table 1, larger privacy budgets lead to more privacy leakage, which allows attackers to infer individual links in the original networks with higher accuracy. While some DP-constrained deep learning models are observed to be sensitive to the sampling ratio during training (Abadi et al., 2016; Shokri & Shmatikov, 2015), the privacy protection of DPGGAN is robust when $q$ is changed in large ranges in practice.

# E   APPENDIX: QUALITATIVE GRAPH VISUALIZATIONS

To understand the behaviors and performances of compared algorithms, we conduct visualizations between the original graphs and graphs generated graphs by different algorithms. We mainly focus on the visualization of DBLP author networks, since they are in general smaller, sparser and thus easier to visually inspect. We draw the graphs with NetworkX[2].

In general, as we can observe in Figure 4-10:

1. The graphs generated by algorithms without DP constraints like NetGAN and GraphRNN are more similar to the original graphs, which is consistent with our results in Table 1 and 2 in the main paper.

2. After enforcing the DP constraints, the graphs generated by DPGVAE are significantly different from the graphs generated by GVAE, especially regarding local structures around individual nodes.

3. The graphs generated by DPGGAN, while also having different local structures compared with those generated by GVAE, have more similar global structures with the original graphs.

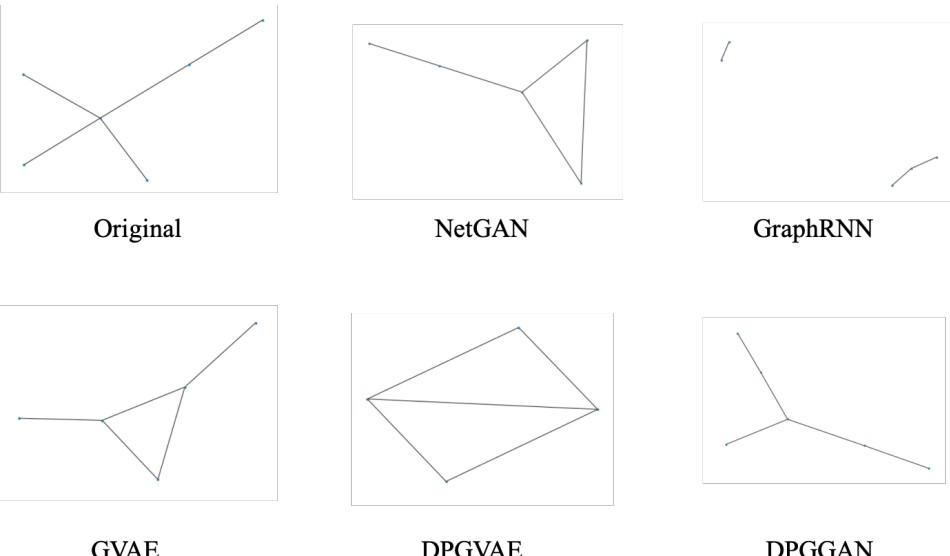

Figure 4: **Visualizations on DBLP author network 1.**

[2]https://networkx.github.io/

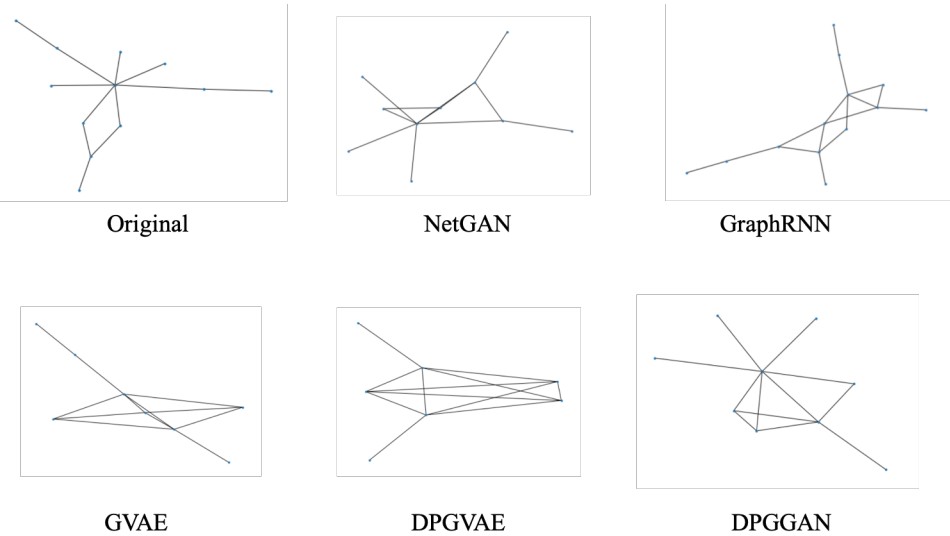

Figure 5: **Visualizations on DBLP author network 2.**

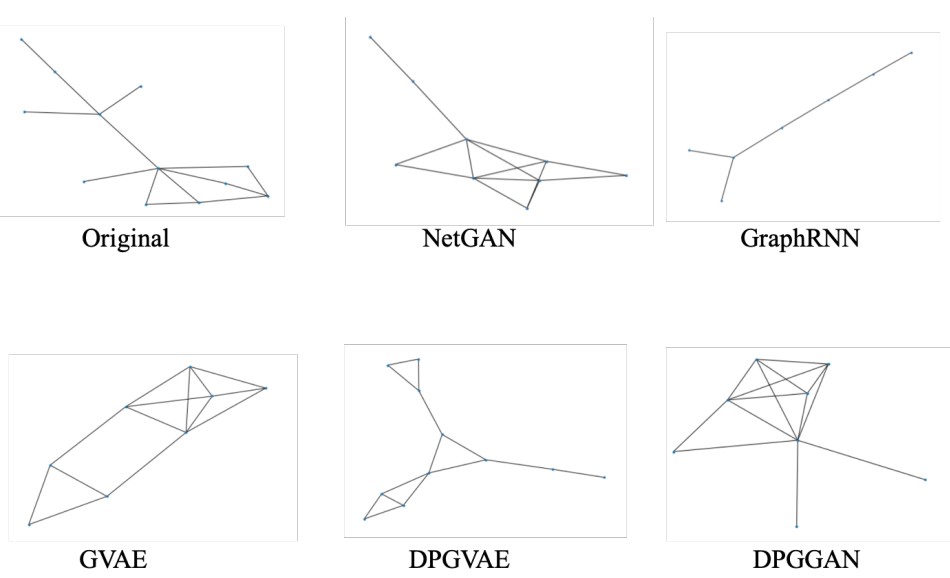

Figure 6: **Visualizations on DBLP author network 3.**

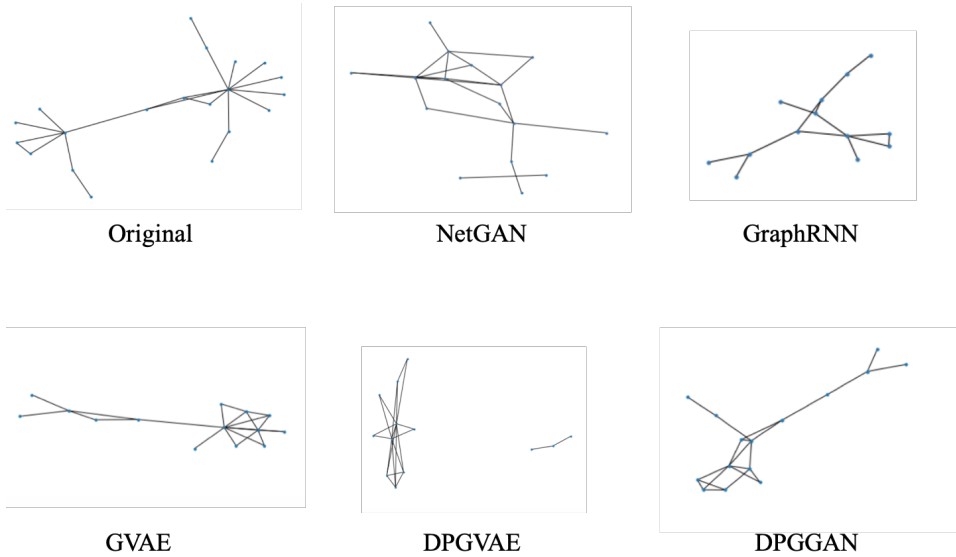

Figure 7: **Visualizations on DBLP author network 4.**

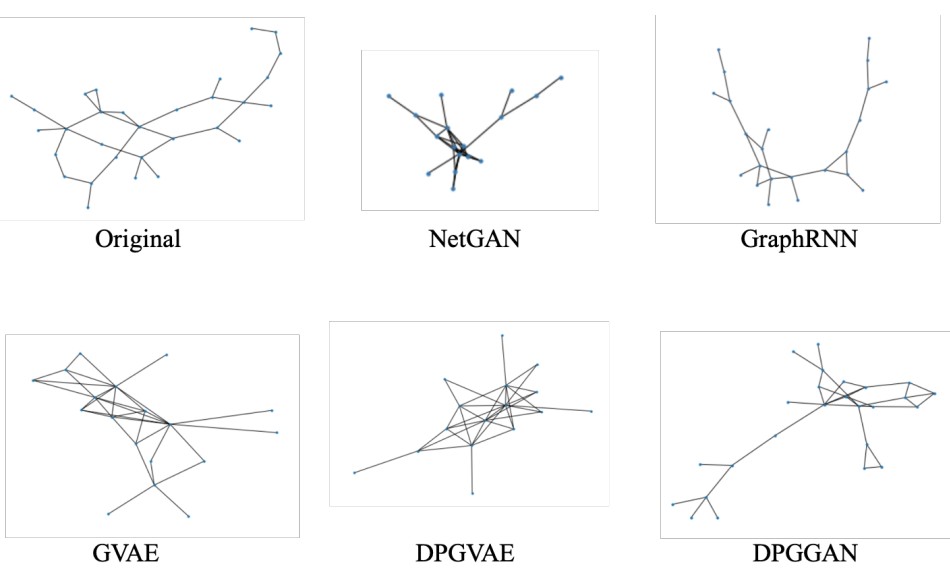

Figure 8: **Visualizations on DBLP author network 5.**

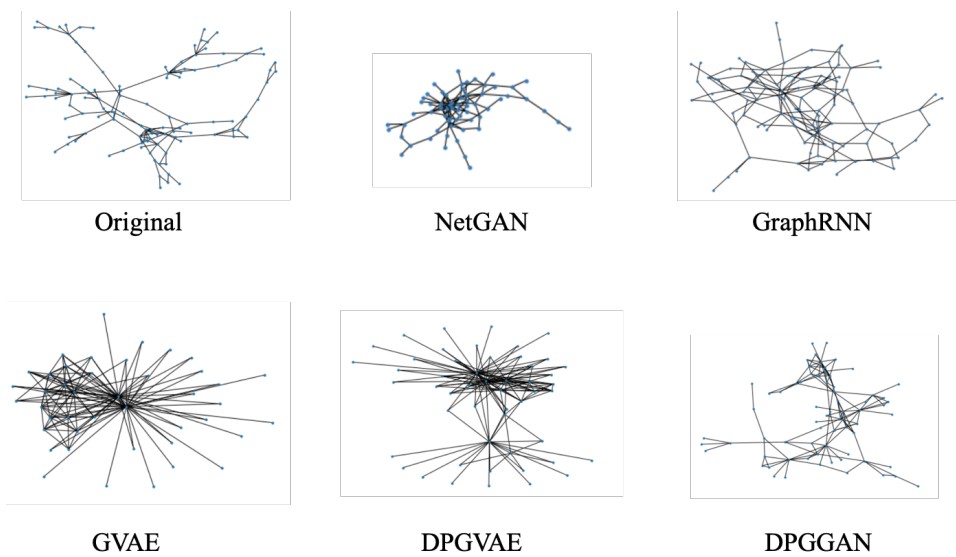

Figure 9: **Visualizations on DBLP author network 7.**

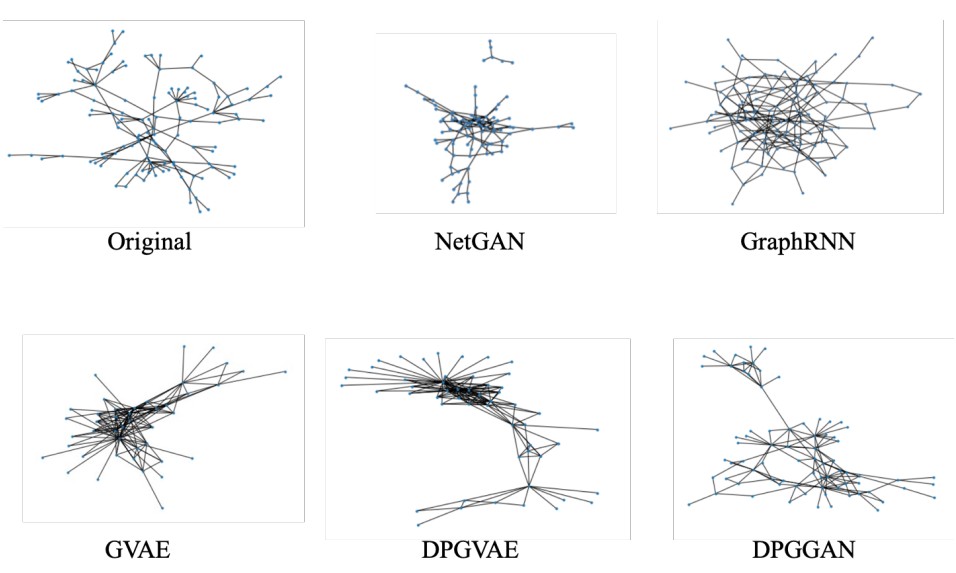

Figure 10: **Visualizations on DBLP author network 7.**

