# OpenReview forum: "Secure Network Release with Link Privacy"
_ICLR.cc/2021/Conference — Reject_

### Official Review · AnonReviewer3 · 2020-10-28

**Rating:** 6
**Confidence:** 4

**Review:**

This work consider the problem of link privacy when releasing models that are trained on graph data. It achieves this by making a generative graph model based on a VAE differentially private. Differential privacy is obtained by adding noise to gradients during training (DPSGD approach). The paper uses graph metrics and a classification downstream task to evaluate the utility of generated graphs. Comparison of algorithms with related work and details of the mechanism could be expanded to articulate novelty and significance of the work.

I. The paper is well written and nicely merges ideas of DP + VAE + graphs for edge privacy.
II. It would be good to see the main body of the paper articulating what the difference is between papers that use DPSGD from Abadi et al. as is and this paper. That is, what exactly had to be changed for graph data. Even after reading the appendix it was not obvious.
III. Experimental results could include accuracy of classification tasks of graph network models as it seems baselines in the paper use non-private generated networks only for downstream tasks. It may suggest that accuracy of generated networks on these tasks is already not optimal and the utility impact of DP is smoothened as a result.

Detailed comments:
1. Theorem 1 and page 5. Please consider adding what is different from DPSGD mechanism; what was the challenge in applying DPSGD in this setting.
2. A2 states "However, in a more complex task like graph learning, a minor change in the training dataset can probably induce a different gap according to the chosen measurement." The use of word "probably" is worrisome. Sensitivity should be properly analyzed when guaranteeing DP and adding noise based on it.
3. Lemma 1: please articulate how s and C depend on each other or relate, if at all. How is this result  different from that in Abadi et al. What would noise addition/clipping be?
4. Theorem 1 proof in Appendix: please explain how this is different from Abadi et al. given that s=C.
5. Does Algorithm 1 make use of s? If not why Lemma 1 was needed.
6. Please explain how Algorithm 1 differs from DPGAN work in Xie et al 2018.

---

> ### Author Response · Authors · 2020-11-14
> **Author Response to Reviewer 4 (Part 1/2)**
>
> We thank the reviewer for the detailed comments.
>
> Q II: It would be good to see the main body of the paper articulating what the difference is between papers that use DPSGD from Abadi et al. as is and this paper. That is, what exactly had to be changed for graph data. Even after reading the appendix it was not obvious.
>
> A II: There are many differences between our work and [Abadi et al], mainly technique-wise and slightly theory-wise. Please refer to [Point 2] and [Point 3] in our major response.
>
> Q III: Experimental results could include accuracy of classification tasks of graph network models as it seems baselines in the paper use non-private generated networks only for downstream tasks. It may suggest that accuracy of generated networks on these tasks is already not optimal and the utility impact of DP is smoothened as a result.
>
> A III: We have compared different models on the down-stream applications of graph classification (Table 2 in our original draft, the GIN acc. columns) and link prediction (Figure 3 on Page 18 in the Appendix). The performance of no-privacy models are indeed pretty strong on both tasks, indicating a thread of too much information contained in the generated graphs (especially from the link prediction results as this directly threatens individual link privacy).
>
> Q1: Theorem 1 and page 5. Please consider adding what is different from DPSGD mechanism; what was the challenge in applying DPSGD in this setting.
>
> A1: As discussed in [Point 2] of our major response, there are basically many possible places to add DPSGD on a rather complicated model like our Graph VAEGAN, and we figured out to just apply it on the gradients of the generator network, so as to minimizing the impact on performance while rigorously guarantee edge-DP. This is discussed in the “Improving structure learning” subsection on Page 6 in our original draft (the paragraph starting with “Following Gu et al”), and we will further highlight it in an updated draft.
>
> Q2: A2 states "However, in a more complex task like graph learning, a minor change in the training dataset can probably induce a different gap according to the chosen measurement." The use of word "probably" is worrisome. Sensitivity should be properly analyzed when guaranteeing DP and adding noise based on it.
>
> A2: We used the word “probably” because DP learning on graph data in a more general setting can potentially be challenging, such as when edge-DP is considered for node classification tasks or node-DP is considered for link prediction tasks (more details discussed in [Point 3] of our major response). For our specific task of ensuring edge-DP with a link reconstruction based graph generation model, the whole derivation happens to be similar to [Abadi et al], which is a benefit from our problem formulation and model design. However, as a theoretical contribution, we provide a more general privacy and sensitivity analysis framework for DP learning on graph data, as the analysis in our Theorem 1 can be adopted in those other graph mining settings (more details discussed in [Point 3] of our major response).
>
> Q3: Lemma 1: please articulate how s and C depend on each other or relate, if at all. How is this result different from that in Abadi et al. What would noise addition/clipping be?
>
> A3: C is a factor controlling the sensitivity s of the gradient function. As C is applied for clipping the gradient value to its norm no more than C, for the graph link reconstruction task, changing one record of input shall at most induce a variation of 1*C of the gradient output (detailed analysis is specified in the proof for Theorem 1 in Appendix A of our original draft). Therefore, the sensitivity s of function f mentioned in Lemma 1 and 2 is mapped to the sensitivity 1*C=C of clipped gradient function, SUM(gradient/max(1,||gradient||_2/C)), in Theorem 1. The noise addition/clipping technique we apply is specified in the Algorithm 1, ie, following the constraints in Theorem 1, we adaptively reduce C with C=gamma*C to maintain a better performance of training while not affecting the privacy loss.

---

> > ### Author Response · Authors · 2020-11-14
> > **Author Response to Reviewer 4 (Part 2/2)**
> >
> > Q4: Theorem 1 proof in Appendix: please explain how this is different from Abadi et al. given that s=C.
> >
> > A4: s=C is obtained from our task-specific analysis under the definition of edge-DP for graph link reconstruction. [Abadi et al] proposed DPSGD for the standard DP protection for the traditional image classification task which needs to be re-considered under the graph learning scenario. The exact proof of our Theorem 1 is very similar to [Abadi et al] given s=C, but it can be different considering other graph learning scenarios, where our proof of Theorem 1 can be easily adopted,  as discussed in more details in [Point 3] of our major response.
> >
> > Q5: Does Algorithm 1 make use of s? If not why Lemma 1 was needed.
> >
> > A5: s is used to refer to the sensitivity of a function f() in Lemma 1 and 2. Later in Theorem 1, f() is specified as the clipped gradient function SUM(gradient/max(1,||gradient||_2/C)), and s is respectively assigned with the value C (explained in A3 above). Lemma 1 is the preparation for Lemma 2, which helps us obtain the upper bound of the moment accountant for the Gaussian Mechanism of a function with its sensitivity as s. Based on the moment accountant properties and Lemma 1, we derive Lemma 2, i.e., the DP protection for iteratively processing the Gaussian Mechanism for f() with its sensitivity as s. As the DPSGD process can be regarded as iteratively processing the Gaussian Mechanism for the clipped gradient function with sensitivity C (explained in A3 above), together with the foundation of Lemma 2 and the definition of edge-DP, we attain Theorem 1.
> >
> > Q6: Please explain how Algorithm 1 differs from DPGAN work in Xie et al 2018.
> >
> > A6: As discussed in [Point 4] in our major response, we gradually reduce the clipping parameter C to allow dynamic perturbation in practice (in Algorithm 1 Line 21), which is explained in our original draft under Algorithm 1 in the Appendix, and we will highlight this major difference in an updated draft.

---

### Official Review · AnonReviewer4 · 2020-10-29
**Does not deliver on its main promise.**

**Rating:** 3
**Confidence:** 5

**Review:**

This paper proposes a method, Differentially Private Graph Generative Nets (DPGGAN), to release graph in way the preserves utility while preserving privacy. This method trains a deep graph generation model in a differentially private manner, by injecting Gaussian noise to the gradients of link reconstruction module, thereby claiming to guarantee edge privacy, while ensuring utility via a structure learning component based on a variational generative adversarial network (GAN) architecture, to enable structure-oriented graph comparison to the original.

The motivation of the work is sound, yet the work does not deliver on its main promise: while the claim is made that link privacy is protected, there is no experiment to justify this claim. While an experiment is mentioned that purportedly does this, the mentioned results are not shown, neither in the main paper, nor in the appendix. The closest the work comes to reporting such results is this statement:

  "... links predicted on the networks generated by DPGGAN are much less accurate than those predicted on the original networks (26%-35% and 15%-20% AUC drops on DBLP and IMDB, respectively) as well as the networks generated by all baselines."

These results are not presented, and the privacy parameters ε that lead to them are not discussed. Thus, it is hard to know how much utility has to be sacrificed for the sake of the privacy gains mentioned. Further, a drop of 15% does not corroborate the claim that the released data useless for link prediction.

Overall, while the paper claims to offer robust privacy guarantees towards various graph attacks, such guarantees are not explicitly spelled out. It seems to be taken for granted that a differentially private link reconstruction provides the intended guarantees, yet that begs the question of what protection is achieved in practice, under a learning-based attack. The fact that learning under differentially privacy can be surprising successful, and thus constitutes an attack on differential privacy, has been established in previous work [1]. This paper should reflect more thoroughly on what privacy means in the proposed setting, how it is shown, and what utility it corresponds to. On the other hand, there have been efforts to define explicit guarantees regarding link reconstruction [2, 3], which this work does not take in consideration.

Incidentally, the paper claims that graphs lack efficient universal representations, citing [Dong et al., 2019]. It is not clear how the cited paper, which studies a form of universal graph representation, supports this statement. Besides, there is work explicitly dealing with efficient universal graph representations [4], based on similar principles to those studied in the cited work.

Last, previous work [5] has already established, to a higher degree that it is done here, that any kind of structural identification attack can effectively be prevented using random edge perturbation, even while important properties of the whole network, as well as of subgraphs thereof, can be accurately calculated on the perturbed data. Given these results, it is unclear what this work adds to what has been already established.

References:
[1] Personal privacy vs population privacy: learning to attack anonymization. KDD 2011.
[2] k-isomorphism: Privacy-preserving network publication against structural attacks. SIGMOD 2010.
[3] L-opacity: Linkage-Aware Graph Anonymization. EDBT 2014.
[4] NetLSD: Hearing the Shape of a Graph. KDD 2018.
[5] Delineating social network data anonymization via random edge perturbation. CIKM 2012.

---

> ### Author Response · Authors · 2020-11-14
> **Author Response to Reviewer 3 (Part 1/2)**
>
> We thank the reviewer for the detailed comments.
>
> Q1: The motivation of the work is sound, yet the work does not deliver on its main promise: while the claim is made that link privacy is protected, there is no experiment to justify this claim. While an experiment is mentioned that purportedly does this, the mentioned results are not shown, neither in the main paper, nor in the appendix. The closest the work comes to reporting such results is this statement:
>
> "... links predicted on the networks generated by DPGGAN are much less accurate than those predicted on the original networks (26%-35% and 15%-20% AUC drops on DBLP and IMDB, respectively) as well as the networks generated by all baselines."
>
> These results are not presented, and the privacy parameters $\epsilon$ that lead to them are not discussed. Thus, it is hard to know how much utility has to be sacrificed for the sake of the privacy gains mentioned. Further, a drop of 15% does not corroborate the claim that the released data useless for link prediction.
>
> A1: We have done experiments to justify the claim on individual link privacy, as discussed in the “Protecting individual links” subsection and the detailed results are already reported in the Appendix (Page 18 Appendix D Figure 3) in our original draft. Moreover, we want to highlight that this task only serves as a proxy to the actual edge-DP, which is rigorously guaranteed by our utilization of DPSGD and DP theory itself.
>
> Q2: Overall, while the paper claims to offer robust privacy guarantees towards various graph attacks, such guarantees are not explicitly spelled out. It seems to be taken for granted that a differentially private link reconstruction provides the intended guarantees, yet that begs the question of what protection is achieved in practice, under a learning-based attack. The fact that learning under differentially privacy can be surprising successful, and thus constitutes an attack on differential privacy, has been established in previous work [1]. This paper should reflect more thoroughly on what privacy means in the proposed setting, how it is shown, and what utility it corresponds to. On the other hand, there have been efforts to define explicit guarantees regarding link reconstruction [2, 3], which this work does not take in consideration.
>
> A2: First, it is NOT taken for granted that our model can guarantee the desired link privacy, but it is rather designed on purpose and fully derived in theory. The whole purpose of utilizing DPSGD and DP theory is to exactly argue that an edge-DP graph generation model can achieve individual link privacy, in the sense that any two graphs differ by one edge can not be distinguished by the model, so as the attacker who has access to any graphs generated by the model. Second, our whole paper is indeed built on the assumption that DP can successfully guarantee data privacy, and particularly edge-DP can guarantee individual link privacy, which has been studied by existing works like [Blocki et al]. There might be some specific corner cases where DP can fail, such as studied in the referenced paper [1]. However, as we assume the success of DP in the first place (so as various existing papers), the discussion of the effectiveness of DP can be interesting but not a necessity. After all, does it mean every single paper that applies DP without such study on possible corner cases is useless? Finally, regarding missing discussions on the referenced papers [2, 3], again this is because we have assumed the success of DP, the whole purpose of which is to relieve exhaustive research from ad hoc definitions of data privacy and their corresponding unprincipled analysis. As we believe DP is a trustable framework to guarantee data privacy, at least in this work, the purpose is to utilize this framework, particularly by applying DPSGD on our designed link reconstruction graph generation model to enforce edge-DP, and use such edge-DP as a guarantee of individual link privacy. As for whether such guarantees shall fail in some special cases, it is a very valid future work.

---

> > ### Author Response · Authors · 2020-11-14
> > **Author Response to Reviewer 3 (Part 2/2)**
> >
> > Q3: Incidentally, the paper claims that graphs lack efficient universal representations, citing [Dong et al., 2019]. It is not clear how the cited paper, which studies a form of universal graph representation, supports this statement. Besides, there is work explicitly dealing with efficient universal graph representations [4], based on similar principles to those studied in the cited work.
> >
> > A3: By claiming graphs lack efficient universal representation, we basically mean there is no way to simply represent a graph as a sequence or tensor like text or image data, which are easier to directly feed into a machine learning model, considering the variable graph sizes and arbitrary node orders, which is discussed in many papers on graph representation and generation including [Dong et al]. There are also works like the referenced paper [4] studying such representations, but additional computations have to be done to get such universal representations. This is the well-recognized challenge in modeling graph data, and it motivates us to design a model that can directly take graph data as input without additional computations such as in [4].
> >
> > Q4: Last, previous work [5] has already established, to a higher degree that it is done here, that any kind of structural identification attack can effectively be prevented using random edge perturbation, even while important properties of the whole network, as well as of subgraphs thereof, can be accurately calculated on the perturbed data. Given these results, it is unclear what this work adds to what has been already established.
> >
> > A4: We would simply be careful to use the word “any”, since we can doubt whether there are failure cases in “any” seemingly wholesome framework, just like how R3 doubts DP can fail in our setting. Step back and say, assuming “any structural attacks” can be prevented by [5] as they claim, there could be other attacks beyond the “structural attacks” as they define, and our focus on individual link privacy can be different from their “structural attacks”. Even one more step back, if the recognition of an individual link as we care about can be exactly formulated as one of their “structural attacks”, we are studying deep graph generation models, while they are studying graph anonymization methods. The two tasks and problem settings are different and even incomparable. One can certainly use their methods to anonymize a graph before feeding it into any deep graph generation model to hopefully generate a private yet utilizable graph, but what if we want to generate a single graph from a set of graphs, generate a set of graphs from a single graph, or generate a set of graphs from a set of graphs? A deep graph generation model can easily generalize to sets of graphs to capture their shared structures and generate an arbitrary number of graphs sharing such captured structures, but what can be preserved if [5] is to be applied individually on a set of graphs?
> >
> > All in all, we appreciate the reviewer to point out several critics on our paper, and we are willing to elaborate more discussions regarding these interesting angles. However, we think many of the requirements are inappropriate and unfair, as they deviate from the main purpose of this work based on a rather common assumption on DP.

---

### Official Review · AnonReviewer2 · 2020-10-29
**Review for paper #130**

**Rating:** 5
**Confidence:** 2

**Review:**

The paper considers the problem of learning graph properties with differential privacy (DP) constraints on edges. The problem is well-motivated in the paper with potential applications in social network learning where interactions between users (nodes) are sensitive.

The techniques involve recent advances in graph generation networks and DP deep learning. The network structure is based on recently proposed Graph VAE and VAEGAN. DPSGD is used during the training process to ensure the privacy guarantee of the process. Experiments are conducted to compare the proposed algorithm to a few nonprivate algorithms and the performance is comparable in certain applications.

An interesting observation of the paper is that it is enough to preserve privacy when training the generator network since the inference is only based on the generated network.

My main concern about the paper is its novelty. It seems to combine results from different areas. I hope the novelty can be further explained in the response.

Other comments:
1. In the experiments, it would be better to report delta values for the privacy guarantees for each experiment.
2. It seems the exact nonprivate counterpart of DPGGAN is not included in the experiments. It would be nice to have results on it to see the exact influence of privacy. For example, in Table 2, Both DPVAE and DPGGAN seem to perform better than NetGAN and GraphRNN algorithms. It would be better if the authors can decouple the influence of privacy and network structure.
3. It seems unnecessary to include the whole proof of DPSGD in the appendix since it is almost identical to the original paper. Including the sensitivity analysis would be enough to me.

---

> ### Author Response · Authors · 2020-11-14
> **Author Response to Reviewer 2**
>
> We thank the reviewer for the detailed comments. Regarding overall novelty, please refer to the detailed discussions in our major response from the four perspectives.
>
> Q1: In the experiments, it would be better to report delta values for the privacy guarantees for each experiment.
>
> A1: As discussed in [Point 4] of our major response, it is a common practice for existing works using DPSGD to fix delta to a small value like 1/n [Abadi et al, Papernot et al (a), Papernot et al (b)], where n is the number of data points. In our experiment, we followed this practice and simply fixed delta to 10e-5 and specified this in both Appendix B and C in our original draft.
>
> Q2: It seems the exact nonprivate counterpart of DPGGAN is not included in the experiments. It would be nice to have results on it to see the exact influence of privacy. For example, in Table 2, Both DPVAE and DPGGAN seem to perform better than NetGAN and GraphRNN algorithms. It would be better if the authors can decouple the influence of privacy and network structure.
>
> A2: As discussed in [Point 4] of our major response, we are running the full experiments on this model and will report the number as soon as possible. We expect the performance of it to be highly competitive towards state-of-the-art graph generation models with no privacy constraint.
>
> Q3: It seems unnecessary to include the whole proof of DPSGD in the appendix since it is almost identical to the original paper. Including the sensitivity analysis would be enough to me.
>
> A3: As discussed in [Point 3] of our major response, we include the full proof in Appendix to be self-contained, and directly show how DPSGD applied on our link reconstruction based graph generation model can guarantee edge-DP. The proof itself is also not identical to the original DPSGD. Specifically, the proof of our Lemma 2 differs from [Abadi et al] in that we generalize the DP protection of iteratively leveraging Gaussian Mechanism for a function with sensitivity s, rather than 1 in the original DPSGD [Abadi et al]. This generalized conclusion in Lemma 2 enables us to explore the DP protection brought by DPSGD for more complicated learning tasks other than traditional image classification as studied in [Abadi et al], so as to be applicable on our graph data. Moreover, through this full derivation, we are able to pay special attention to some seemingly unimportant parameters like $c_1$ and $c_2$ as specified in our Theorem 1. [Abadi et al] only mentioned $c_1$ and $c_2$ with “There exist constants …” in their Theorem 1 without further specification even in their Appendix. However, $c_1$ and $c_2$ obviously influence the computation of $\epsilon$ and $\sigma$ in DP, so we specifically analyzed them in our Theorem 1 to retrieve a more concrete privacy bound of the outcome model. Finally, we provide this proof as a generic one for DP learning on graph data, where the sensitivity analysis of our task happens to be rather similar to [Abadi et al] due to our problem formulation and model choice, but in other scenarios such as when edge-DP is considered for node classification tasks or node-DP is considered for link prediction tasks, we can basically follow the similar analysis in our Theorem 1 to derive different results accordingly (details discussed in [Point 3] of our major response). We will properly highlight these points in an updated draft.

---

### Official Review · AnonReviewer1 · 2020-11-02
**Secure Network Release with Link Privacy**

**Rating:** 6
**Confidence:** 3

**Review:**

This paper considers the problem of releasing sensible structured data, where there are two inlined challenges: 1. The global network structure should be effectively preserved; 2. The link privacy should be rigorously protected. This paper looks at the secure release of network data with deep generative models. Specifically, the paper develops two models, DPGVAE and DPGGan, which can be viewed as a combination of DP-SGD and graph generation techniques. Extensive experiments are carried out on real-world network datasets, and the positive results have shown the effectiveness of the new models.

I think the problem this paper looks at is both interesting and fundamental. Most of the current DP works focus on estimating one specific property of the dataset's underlying distribution, e.g., mean of the distribution. However, in many real-world applications, the tasks can not be known beforehand, and a potential solution is to release synthetic datasets under DP constraint. As far as I know, there are few papers in this area of privately releasing synthetic datasets with deep generative models.

This paper has proposed two models (DPGVAE and DPGGan), both reasonable to me. Meanwhile, experiments are carried out on real-world network datasets, and it has shown that for many structural statistics, the algorithms have competitive performance. I have little background knowledge of graph generation, and I can not give many technical comments:
1. I will be appreciated if the authors can provide the results when $\varepsilon = \infty$, which I expect to be better than the case when $\varepsilon = 10$. Does it mean your algorithm outperforms the current best algorithm in IMDB when there is no privacy constraint?
2. I think it is also interesting to include some experiments on synthetic datasets. I believe for some extreme graph structure, DP should have more impact. For example, if the objective is to estimate the number of triangles in the graph, I expect the privacy to incur more loss when the underlying graph is complete compared with a sparse graph.
3. From the algorithmic perspective, I  think both models can be viewed as a simple combination of DP-SGD and graph generation techniques. I am not sure this paper has made many theoretic contributions.

---

> ### Author Response · Authors · 2020-11-14
> **Author Response to Reviewer 1**
>
> We thank the reviewer for the detailed comments.
>
> Q1: I will be appreciated if the authors can provide the results when epsilon=infinity, which I expect to be better than the case when epsilon=10. Does it mean your algorithm outperforms the current best algorithm in IMDB when there is no privacy constraint?
>
> A1: Regarding DPGGan with epsilon = infinity, we are running the full experiments on this model and will report the number as soon as possible. We expect the performance of this model to be highly competitive towards state-of-the-art graph generation models with no privacy constraint (and even outperforming some of them, due to our novel design of Graph VAEGAN).
>
> Q2: I think it is also interesting to include some experiments on synthetic datasets. I believe for some extreme graph structure, DP should have more impact. For example, if the objective is to estimate the number of triangles in the graph, I expect the privacy to incur more loss when the underlying graph is complete compared with a sparse graph.
>
> A2: As discussed in [Point 4] in our major response, this is a good suggestion, and we will try to give more interesting results on this in our final version. However, compared to synthetic experiments, we have verified our approach on two widely used real-world datasets. The used datasets include graphs with various link density (0.0019-0.1250 for DBLP and 0.0616-0.4868 for IMDB) and other structural properties.
>
> Q3: From the algorithmic perspective, I think both models can be viewed as a simple combination of DP-SGD and graph generation techniques. I am not sure this paper has made many theoretic contributions.
>
> A3: In this work, we give a new extension on the theoretical analysis of DPSGD on graph data as discussed in [Point 3] of our major response. We will point this out clearly in an updated draft. However, theoretical contribution is our major claim or only focus in this work. As discussed in [Point 2] of our major response, the proposed models may seem straightforward, but we have gone through great efforts to figure out the detailed designs from many possible solutions. Thus we think the technical contribution itself is also significant.

---

### Author Response · Authors · 2020-11-14
**Major response to all reviewers: regarding our main novelty and contributions (Point 1-2)**

We thank all reviewers for their time and effort in providing the valuable feedback.
Here we would like to first highlight the main contributions in this work as follows:

[Point 1] Problem-wise: we are the first to consider the secure generation of network data with deep learning models. We find a natural use case of DP theory in network generation, which is the preservation of global network data utility and local link privacy. Although seemingly straightforward, no previous work has ever considered such a setting. Yet it is indeed an important and practical setting.

[Point 2] Technique-wise: we propose to train the GraphVAE model with DPSGD and further improve the model through VAEGAN. Again, this framework may look straightforward, but there are many other graph generation models, among which we find GraphVAE to be the most proper backbone, due to our insight that such a link reconstruction based graph generation model can be properly trained with DPSGD to guarantee link privacy without severely damaging global network structure. Moreover, we find VAEGAN to be a proper extension to DPGVAE, since it does not change the link reconstruction objective but enforces better graph structure learning. Finally, other than DPSGD, there are many frameworks to ensure data privacy, such as the ones mentioned by R4 [2, 3, 5]. However, without the support of DP theory, these works have no principled theoretical guarantee towards data privacy, but rather can only demonstrate certain types of privacy regarding specific empirical measures. Even considering DP, there are many possible options regarding where and how to add the noises (eg, the noises can be added to training data itself, hidden representation after the encoder of GraphVAE, samples as input of the decoder of GraphVAE, the graph discriminator of VAEGAN, etc). We have considered all such options and conducted empirical studies to find them either severely damaging the generated graph structures or impossible to be rigorously bounded regarding limited privacy budgets. Even considering DPSGD, applying it on the complicated model of Graph VAEGAN is not trivial. We figured it out through rigorous thinking and empirical trial-and-errors that it is the most appropriate (with least influence on the performance but best DP guarantee) to only perturb the gradients of the generator of Graph VAEGAN, and we gradually reduce the clipping parameter C to allow dynamic perturbation in practice (in Algorithm 1 Line 21 as different to Xie et al 2018 asked by R4), achieving the best performance without worsening the DP guarantee. Due to concise presentation, we did not extend on all such model choices in the paper, but rather just presented the best model we came up with and experimental results regarding the major variants. This does not mean there is less effort in developing the technique, and we will further highlight some of the points above in an updated version.

---

> ### Author Response · Authors · 2020-11-14
> **Major response to all reviewers: regarding our main novelty and contributions (Point 3)**
>
> [Point 3] Theory-wise: as noticed by the reviewers, our utilization of DPSGD to the particular model of Graph VAEGAN does not require significantly different theoretical justification as those in [Abadi et al].  This is a major benefit of our appropriate problem setting and technical design. Nonetheless, we provide the full derivation of DP guarantee regarding our specific application of DPSGD on our Graph VAEGAN model, regarding parameter setting, sensitivity, etc. We include such a similar proof to that of [Abadi et al], because we want to be self-contained and exactly show why the direct application of DPSGD on this model can achieve the claimed edge-DP. Through such detailed derivations, we are also able to pay special attention to some seemingly unimportant parameters like $c_1$ and $c_2$ as we specified in our Theorem 1, which are bypassed in [Abadi et al] and other existing works assuming everything is simply the same as [Abadi et al] when DPSGD is applied. However, $c_1$ and $c_2$ obviously influence the computation of $\epsilon$ and $\sigma$ in DP, and should be specified to retrieve a more concrete bound of the exact outcome model as shown in our Theorem 1. Finally, this derivation also serves as a general DP analysis when DPSGD is considered on graph data, which covers our problem as a specific instance appearing to be rather similar to [Abadi et al], because we care about edge-DP and our graph generation model is based on link reconstruction. However, there are many other settings of learning with graph data. With similar analysis of the proof for Theorem 1, we can conduct edge-DP results for other graph models trained with the same technique. For example, for a generative graph model solving the node classification task, changing 1 edge in the input graph shall affect 2 nodes' representations. Thus, when the gradient value is clipped with C, the sensitivity for the node classification model’s clipped gradient function is now 2*C. Moreover, regarding node-DP [Kasiviswanathan et al., 2013] instead of edge-DP, one can also deliver the corresponding sensitivity analysis following our analysis here. For example, for the undirected graph with N nodes without duplicated links nor links starting and ending at the same node, when 1 node in the input graph changes, the respective gradient function clipped with C for training the link reconstruction model shows at most (N-1)*C difference. Therefore, under the settings of our Theorem 1, when it comes to node-DP, the respective sensitivity of the clipped gradient function for a link reconstruction model is (N-1)*C, which is much larger than that for edge-DP. Beyond such purposes, we do not intentionally attempt to claim significant theoretical contribution in this work. Instead, more rigorous study and analysis of DP theory as well as its empirical guarantees on complex deep learning frameworks like GAN (such as what does privacy mean in a DP setting as suggested by R3) is a rather open problem, which is beyond the purpose of this work. After all, many impactful papers directly using DPSGD without such analysis have been published and well-cited such as [Papernot et al (a), Papernot et al (b), Jordon et al, Li et al]. Therefore, we do not think this is an appropriate expectation of our work or a fair reason for rejection.

---

> > ### Author Response · Authors · 2020-11-14
> > **Major response to all reviewers: regarding our main novelty and contributions (Point 4)**
> >
> > [Point 4] Experiment-wise: we may have mentioned some of this in [Point 2] and [Point 3] already. We want to further highlight that we have done extensive experiments on two standard graph generation datasets, directly showing that our proposed model is superior in both global structure preservation and individual link protection. Specifically, the global structure preservation capability of our model is very competitive compared with state-of-the-art graph generation models (we did miss out the results on a DPGGan model without privacy constraint and we are running the full experiments on that and will report the numbers as requested by R1 and R2 ASAP). R1 further suggested to add experiments with synthetic graphs regarding certain graph properties like link density. We do not think this is very necessary or of top priority right now, since the real-world datasets we use now already include graphs of varying link density (0.0019-0.1250 for DBLP and 0.0616-0.4868 for IMDB) and other graph properties as shown in Table 1&2. We will add more experiments on synthetic graphs if time permits. Regarding the delta values asked by R2, it is a common practice for existing works using DPSGD to fix delta to a small value like 1/n, where n is the number of data points. In our experiment, we followed this practice and set delta to 10e-5 and specified this in both Appendix B and C in our original draft. On the link privacy part, R3 asks us about the link prediction results and thinks it is missing even in the Appendix, but it is actually right in the Appendix of our original draft (Page 18 Appendix D Figure 3). We also pointed this out in the main content (Page 8 at the end of the subsection of Protecting individual links), which is rather hard to miss. The two figures in Figure 3 clearly show DPGGan to have a worse performance in reversed link prediction compared with all other algorithms, which serves as a proxy to the rigorous edge-DP guarantee. Again, we utilize DP theory to rigorously guarantee edge-DP, instead of basically relying on such proxies to argue about data privacy, which is essentially different from [2, 3, 5] as mentioned by R3.
> >
> > References
> >
> > [Abadi et al] Deep learning with differential privacy. SIGSAC 2016.
> >
> > [Papernot et al (a)] Semi-supervised knowledge transfer for deep learning from private training data. ICLR 2017.
> >
> > [Papernot et al (b)] Scalable private learning with pate. ICLR 2018.
> >
> > [Jordon et al] Pate-gan: generating synthetic data with differential privacy guarantees. ICLR 2019.
> >
> > [Li et al] Differentially private meta-learning. ICLR 2020.
> >
> > [Acs et al] Differentially private mixture of generative neural networks. TKDE 2019.
> >
> > [Kasiviswanathan et al] Analyzing graphs with node differential privacy. TCC 2013.
> >
> > [Blocki et al] The johnson-lindenstrauss transform itself preserves differential privacy. FOCS, 2012.

---

### Decision · Program_Chairs · 2021-01-07
**Final Decision**

**Decision:**

Reject

**Comment:**

This paper studies synthetic data generation for graphs under the constraint of edge differential privacy. There were a number of concerns/topics of discussions, which we consider separately:
1. Theoretical contributions. There are not that many theoretical contributions in this paper. I think this is OK, if the other components are compelling enough. On the theory, the authors mention that accounting for the constants is important in the analysis of DPSGD. On the contrary, I would say that these constants are not very important: if one requires specific constants, numerical procedures can determine values, otherwise for the sake of theory, no one generally needs these constant factors.

2. Empirical/experimental contributions. This was the primary axis for evaluation for this paper. None of the authors were especially compelled by the results. The methods are essentially combinations of known tools from the literature, and it is not clear why these are the right ones to solve this problem in particular. If the results were very exciting, that might be sufficient to warrant acceptance, but it is still not clear how significant the cost of privacy is in this setting. The experiments are not thorough enough to give serious insight here. It is a significant oversight to not provide results on DPGGAN without the privacy constraint, as this is the best performing model with privacy. The omission of something as important as this (and lack of inclusion in the response, with only a promise to include later) is indication that the experiments are not sufficiently mature to warrant publication at this time. The decision of rejection is primarily based on concerns related to the empirical and experimental contributions.

3. Privacy versus link reconstruction. Reviewer 4 had concerns about the notion of privacy, claiming that it does not correspond to the probability of a link being irrecoverable. This is differential privacy "working as intended", which is not intended to make each link be irrecoverable: it is simply to make sure the answer would be similar whether or not the edge were actually present, so it may be possible to predict the presence of an edge even if we are differentially private with respect to it (e.g., the presence of many other short paths between two nodes are likely to imply presence of an edge). Some discussion of this apparent contradiction might be warranted, as this might mislead reader who are specifically trying to prevent edge recovery. It might also be worthwhile to have discussion of node DP in the final paper. The authors comment "we focus on edge privacy because it is essential for the protection of object interactions unique for network data compared with other types of data" -- the stronger notion of node differential privacy might also be applicable here. It would indeed be interesting to know whether it can preserve the relevant statistics (some of which seem more "global" and thus preservable via node DP).